# Synaptic Interaction Penalty: Appropriate Penalty Term for Energy-Efficient Spiking Neural Networks

**Kazuma Suetake**\*                                        *kazuma.suetake@aisin-software.com*
*AISIN SOFTWARE, Aichi, Japan*

**Takuya Ushimaru**\*                                        *takuya.ushimaru@aisin-software.com*
*AISIN SOFTWARE, Aichi, Japan*

**Ryuji Saiin**                                        *ryuji.saiin@aisin-software.com*
*AISIN SOFTWARE, Aichi, Japan*

**Yoshihide Sawada**                                        *yoshihide.sawada@aisin.co.jp*
*Tokyo Research Center, AISIN, Tokyo, Japan*

**Reviewed on OpenReview:** *https://openreview.net/forum?id=42BKnT2qW3*

## Abstract

Spiking neural networks (SNNs) are energy-efficient neural networks because of their spiking nature. However, as the spike firing rate of SNNs increases, the energy consumption does as well, and thus, the advantage of SNNs diminishes. Here, we tackle this problem by introducing a novel penalty term for the spiking activity into the objective function in the training phase. Our method is designed so as to optimize the energy consumption metric directly without modifying the network architecture. Therefore, the proposed method can reduce the energy consumption more than other methods while maintaining the accuracy. We conducted experiments for image classification tasks, and the results indicate the effectiveness of the proposed method, which mitigates the dilemma of the energy–accuracy trade-off.

## 1 Introduction

With the rapid growth and spread of neural networks, realizing energy-efficient neural networks is an urgent mission for sustainable development. One such model is the spiking neural network (SNN), which is also known to be more biologically plausible than ordinary artificial neural networks (ANNs). SNNs are energy-efficiently driven on neuromorphic chips (Akopyan et al., 2015; Davies et al., 2018) or certain field-programmable gate arrays (FPGAs) (Maguire et al., 2007; Misra & Saha, 2010) by asynchronously processing spike signals. However, as the spike firing rate of an SNN increases, the energy consumption does as well, and thus, the advantage of the SNN diminishes. Therefore, in addition to the shift from ANNs to SNNs, it is advantageous to adopt training methods that reduce energy consumption in the inference phase. At the same time, such a training method should be independent of the network architecture to avoid limitations in the application. That is, our goal is to develop a training method that realizes energy-efficient SNNs without any constraint on the network architecture.

There are various approaches toward energy-efficient SNNs, such as pruning, quantization, and knowledge distillation (Kundu et al., 2021; Chowdhury et al., 2021a; Lee et al., 2021), which are widely-used approaches also in ANNs. Further, there are SNN-specific approaches sparsifying the spiking activity related to the energy consumption (Lee et al., 2020; Kim & Panda, 2021; Naya et al., 2021), to which our method belongs. In particular, the methods that penalize the spike firing rate in the training phase are close to our aforementioned

---

\*Equal contribution.

goal (Esser et al., 2016; Sorbaro et al., 2020; Pellegrini et al., 2021). However, they indirectly reduce the energy consumption by arbitrarily reducing the spike firing rate, where there is no strict proportionality between them. Hence, reducing energy consumption while maintaining accuracy is difficult.

**Principle and Idea**   Our principle is that—we should optimize the metric as it is in the training phase. In this spirit, we propose to introduce a proper penalty term for the spiking activity—a *synaptic interaction penalty*—into the objective function. It is derived so that its expected value is precisely proportional to the energy consumption metric for the SNN. Although the difference between the proposed and existing methods is only at the scaling factor for each spike, we demonstrate that this minor correction causes significant improvement.

**Main Contributions**

- We derived a novel penalty term that can directly optimize the metric for the total energy consumption of an SNN without modifying the network architecture.

- We demonstrated that the proposed method can reduce the energy consumption more than other methods while maintaining the accuracy for image classification tasks, which mitigates the dilemma of the energy–accuracy trade-off.

- We also demonstrated that the proposed method is compatible with the weight decay, which imposes implicit sparsity on the network (Yaguchi et al., 2018), and that the proposed method creates a higher sparsification effect than the weight decay.

## 2   Related Work

### 2.1   Spike Sparsification in Direct SNN Training

The most relevant approaches to our proposal introduce the penalty term for the spike firing rate and directly train SNNs by the surrogate gradient method (Esser et al., 2016; Pellegrini et al., 2021). It is a straightforward idea to penalize the spike firing rate to obtain energy-efficient SNNs because the spike firing rate appears in the SNN energy consumption metric (Lee et al., 2020; Kim & Panda, 2021). We refer to the reduction in spike firing rate as spike sparsification. Although the spike firing rate cannot be optimized by the ordinal backpropagation method owing to non-differentiability, it can be optimized by the surrogate gradient method, which is the same technique as training spiking neurons of SNNs (Zenke & Ganguli, 2018; Shrestha & Orchard, 2018). However, neither of these penalty terms (Esser et al. (2016); Pellegrini et al. (2021)) precisely matches the energy consumption metric. As opposed to them, our synaptic interaction penalty resolves this limitation.

### 2.2   Spike Sparsification via Conversion from ANN

Other approaches introduce the penalty term for corresponding ReLU networks (ANNs with ReLU activations) and convert them to SNNs (Sorbaro et al., 2020; Narduzzi et al., 2022). Although there is no guarantee that the penalty terms for ReLU networks contribute to the reduction of the energy consumption for converted SNNs, ReLU networks can be optimized by the ordinal backpropagation method. Note that the same synaptic scaling factor for the penalty term as ours is proposed to reduce the energy consumption for SNNs in Sorbaro et al. (2020). However, they failed to provide evidence to support their claim, as they mentioned. As opposed to them, we provide theoretical and experimental proof in the setting of the direct SNN training by the surrogate gradient.

### 2.3   Neuron Sparsification

Neuron sparsification means increasing the number of permanently zero-valued activations for all data—dead neurons. It is a stronger condition than spike sparsification, which does not force neurons to be permanently inactive. In ReLU networks, the training with the Adam optimizer and weight decay regularization implicitly

induce neuron sparsification (Yaguchi et al., 2018) because ReLU activations have an inactive state. However, this claim has yet to be demonstrated in the context of SNNs, where spiking neurons also have an inactive state, even though weight decay is usually adopted in SNNs. Therefore, to detect the effect of our method correctly, we shall also focus on the weight decay.

## 3 Method

In this section, we propose the synaptic interaction penalty. First, we describe the spiking neuron model with surrogate gradient mechanism. Next, we describe the metric for energy consumption, which can be represented by the spiking activity. Note that we need to optimize both the accuracy and energy efficiency in the training phase. Finally, we state that the synaptic interaction penalty is the proper penalty term to optimize the energy consumption metric.

### 3.1 Neuron Model and Surrogate Gradient

In this study, we use SNNs constructed by single-step spiking neurons, which are superior to the multi-time step SNNs in terms of training and inference costs for static tasks (Suetake et al., 2023). Note that the single-step spiking neurons are the same setup as that in a previous study of the penalty term (Esser et al., 2016).

Let us denote $l \in \{l \in \mathbf{Z} \mid 1 \leq l \leq L\}$ as the layer, $d_l$ as the number of neurons in the $l$-th layer, $\mathbf{s}_0 = x \in X \subset \mathbf{R}^{d_0}$ as the input data, and the subscript $i$ of any vector as its $i$-th component. Then, the single-step spiking neuron is defined as follows (Suetake et al., 2023).

**Definition 3.1.** The forward mode of a single-step spiking neuron consists of two ingredients: the membrane potential $\mathbf{u}_l \in \mathbf{R}^{d_l}$ and spikes emitted by neurons $\mathbf{s}_l \in \{0, 1\}^{d_l}$. They are defined using the Heaviside step function $H$ as follows:

$$\mathbf{u}_l := \mathbf{W}_l \mathbf{s}_{l-1}, \tag{1}$$

$$s_{l,i}(u_{l,i}) := H(u_{l,i} - u_{\text{th}}) = \begin{cases} 1 & (u_{l,i} \geq u_{\text{th}}), \\ 0 & (u_{l,i} < u_{\text{th}}), \end{cases} \tag{2}$$

where $\mathbf{W}_l \in \mathbf{R}^{d_l \times d_{l-1}}$ is the strength of the synapse connections, also called the weight matrix, and $u_{\text{th}} \in \mathbf{R}$ is the spike firing threshold (Eq. 1 for $l = 1$ corresponds to the direct encoding (Rueckauer et al., 2017)). In this context, we classify the $i$-th neuron in the $l$-th layer as a *dead neuron* if $s_{l,i} = 0$ for all input data within a given dataset.

A backward mode of the single-step neuron as it is does not work in the standard backpropagation algorithm because the derivative of Eq. 2 vanishes almost everywhere. Therefore, we adopt the technique called surrogate gradient, *i.e.,* we formally replace the derivative function with some reasonable function, for example, the following one (Suetake et al., 2023):

$$\frac{\partial s_{l,i}}{\partial u_{l,i}}(u_{l,i}) :\simeq \begin{cases} \frac{1}{\tau}\frac{1}{u_{l,i}} & (u_{l,i} \geq u_{\text{th}}), \\ \frac{\partial \sigma_\alpha}{\partial u_{l,i}}(u_{l,i}) & (u_{l,i} < u_{\text{th}}), \end{cases} \tag{3}$$

where $\tau$ and $\alpha$ are hyperparameters and $\sigma_\alpha$ is the scaled sigmoid function expressed as follows:

$$\sigma_\alpha(u_{l,i}) := \frac{1}{1 + \exp((-u_{l,i} + u_{\text{th}})/\alpha)}, \tag{4}$$

$$\frac{\partial \sigma_\alpha}{\partial u_{l,i}}(u_{l,i}) = \frac{1}{\alpha}\sigma_\alpha(u_{l,i})(1 - \sigma_\alpha(u_{l,i})). \tag{5}$$

Note that the choice of a function for the surrogate function is irrelevant to our proposal.

### 3.2 Metric for Energy Consumption

We prepare the symbol $\psi_{l,i}$ for the number of synapses outgoing from the $i$-th neuron in the $l$-th layer, *i.e.*, the number of matrix elements in $(\boldsymbol{W}_{l+1})_{*,i} \in \boldsymbol{R}^{d_l}$ that is not forced to vanish in terms of network architecture. Let us denote $W_l$ and $H_l$ as the width and height of the feature map, respectively, $C_l$ as the channel size in the $l$-th layer, and $k_{l+1}$ as the kernel size associated with $\boldsymbol{W}_{l+1}$. We restrict both the kernel width and height to be identical to $k_{l+1}$ for the sake of simplicity. Then, explicit forms of $\psi_{l,i}$ are as follows, *e.g.*, the standard fully connected (fc) and two-dimensional convolutional (conv) layers,

$$\psi_{l,i} = \psi_l = C_{l+1} \qquad \text{(fc)}, \tag{6}$$

$$\psi_{l,i} \simeq \psi_l = \frac{W_{l+1}H_{l+1}}{W_l H_l} C_{l+1} k_{l+1}^2 \quad \text{(conv)}, \tag{7}$$

where the convolutional layer assumes appropriate padding with respect to $k_{l+1}$ to satisfy Eq. 7. Additionally, Eq. 7 provides the average value in the case of downsampling layers ($W_{l+1}H_{l+1} < W_l H_l$) because the precise value varies depending on the neuron's position and the downsampling method. Note that $\psi_{l,i}$ may also vary depending on the neuron's position within the convolutional layers, *e.g.*, corners, edges, and other positions, contingent upon hardware implementation. For instance, the hardware design could involve computing all kernels tied to spike firing at edge positions including padding and then discarding redundant computation results (Kang et al., 2020). Alternatively, the hardware could be engineered to evade unnecessary computations in the initial phase through a conditional branch (Bamberg et al., 2023). The expression presented in Eq. 7 remains consistent with the approach adopted by Kang et al. (2020), thereby providing a sound measure that accurately accounts for hardware realization. In addition, it is worth noting that more complex scenarios involving nontrivial padding, stride, and dilatation might introduce additional $i$-dependency in $\psi_{l,i}$. However, such specific cases are beyond the scope of this study.

Using $\psi_{l,i}$, we can express the number of floating point operations (FLOPs), which is often used as a metric to measure the computational complexity in ANNs, as follows:

$$\text{FLOPs}(l) := \sum_{i=1}^{d_l} \psi_{l,i}, \tag{8}$$

and the layer-wise and balanced spike firing rates, which are also important metrics to measure the sparsity of spiking activity in SNNs, as follows:

$$R(l) := \underset{x \in X}{\mathbb{E}} \left[ \frac{\sum_{i=1}^{d_l} s_{l,i}}{d_l} \right], \tag{9}$$

$$R := \frac{1}{L} \sum_{l=1}^{L} R(l), \tag{10}$$

where the operation $\mathbb{E}_{x \in X}$ means taking the empirical expectation in the dataset $X$. Then, the energy consumption metric that we should optimize is defined as follows.

**Definition 3.2.** Let us denote $T$ as the size of time steps and $E_{\text{AC}}$ [pJ] as the energy consumption per accumulate operation. Then, the layer-wise and total energy consumption metrics for the SNN are defined as follows:

$$E_{\text{SNN}}(l) := T E_{\text{AC}} \underset{x \in X}{\mathbb{E}} \left[ \sum_{i=1}^{d_l} \psi_{l,i} s_{l,i} \right], \tag{11}$$

$$E_{\text{SNN}} := \sum_{l=1}^{L} E_{\text{SNN}}(l). \tag{12}$$

Note that $T$ is equal to one for the single-step neuron model.

This total energy consumption metric holds practical validity, as it aligns with the achievable energy consumption levels when implementing SNNs through specific neuromorphic chips. For example, in Kim & Panda (2021); Esser et al. (2016), the implementation of SNNs on the TrueNorth platform was demonstrated (Akopyan et al., 2015), where the energy proportional to $\psi_{l,i}$ is exclusively consumed upon the occurrence of the corresponding spike firing. However, note that we consider the ideal setting for SNN inference, where peripheral energy consumption (Lemaire et al., 2022) does not contribute except for the spiking interaction.

If $\psi_{l,i}$ is independent of $i$ ($\exists \psi_l, \forall i, \psi_{l,i} = \psi_l$), by combining Eqs. 8 and 9, Eq. 11 is rewritten as follows:

$$
\begin{aligned}
E_{\text{SNN}}(l) &= TE_{\text{AC}} \sum_{i=1}^{d_l} \psi_l \; \mathbb{E}_{x \in X} \left[ \frac{\sum_{i=1}^{d_l} s_{l,i}}{d_l} \right] \\
&= TE_{\text{AC}} \text{FLOPs}(l) R(l),
\end{aligned}
\tag{13}
$$

which is the same metric as that used in Kim & Panda (2021).

Note that dead neurons (Def. 3.1) for a given dataset never contribute to the energy consumption metric (Def. 3.2) for that dataset. Moreover, these dead neurons never influence model outputs for the same dataset. Therefore, pruning dead neurons after training can help save memory capacity, provided that data distributions do not differ significantly between before and after training.

### 3.3 Synaptic Interaction Penalty

To optimize the energy consumption $E_{\text{SNN}}$ (Eq. 12), we propose the following penalty terms.

**Definition 3.3.** The layer-wise and total synaptic interaction penalty terms are defined as follows:

$$
\Omega_{\text{syn}}(l) = \Omega_{\text{syn}}(l, \boldsymbol{s}_l) := \frac{1}{p} \sum_{i=1}^{d_l} \psi_{l,i} s_{l,i}^p,
\tag{14}
$$

$$
\Omega_{\text{syn}} = \Omega_{\text{syn}}(\boldsymbol{s}) := \sum_{l=1}^{L} \Omega_{\text{syn}}(l, \boldsymbol{s}_l),
\tag{15}
$$

where $\boldsymbol{s} := \{\boldsymbol{s}_l\}_{l=1}^{L}$ and $p \geq 1$.

The equivalency between the total energy consumption metric and total synaptic interaction penalty immediately follows from their definitions and the equation $s_{l,i}^p = s_{l,i}$, which is derived from Eq. 2.

**Theorem 3.4.** *The expected value of the layer-wise and total synaptic interaction penalties are precisely proportional to the layer-wise and total energy consumption metrics of SNNs:*

$$
pTE_{AC} \; \mathbb{E}_{x \in X} \left[ \Omega_{\text{syn}}(l) \right] = E_{SNN}(l),
\tag{16}
$$

$$
pTE_{AC} \; \mathbb{E}_{x \in X} \left[ \Omega_{\text{syn}} \right] = E_{SNN},
\tag{17}
$$

*for arbitrary $p \geq 1$.*

This fact means that optimizing Eq. 15 leads to optimizing Eq. 12. Hence, we strongly propose to use Eq. 15 as the penalty term to optimize the energy consumption metric. In the following, we indicate the total synaptic interaction penalty when simply referred to as the *synaptic interaction penalty*. The spike firing rate and energy consumption metric represent the balanced spike firing rate and total energy consumption metric, respectively, as well as the synaptic interaction penalty.

Table 1: Comparison among total penalty terms. Our penalty term $\Omega_{\text{syn}}$ is precisely equal to the ground truth $E_{\text{SNN}}/E_{\text{AC}}$.

| Model | $E_{\text{SNN}}/E_{\text{AC}}$ | $\Omega_{\text{syn}}$ | $\Omega_{\text{total}}$ | $\Omega_{\text{balance}}$ |
|---|---|---|---|---|
| CNN7 | 98895888 | 98895888 | 59688 | 6 |
| VGG11 | 2526060544 | 2526060544 | 249856 | 10 |
| ResNet18 | 553730048 | 553730048 | 671744 | 20 |

*Remark* 3.5. The proposed penalty term can be optimized in the manner of the surrogate gradient as in Sec. 3.1, and using $p \neq 1$ options controls the backward signal when the spike does not fire as follows:

$$\frac{1}{p}\frac{\partial s_{l,i}^p}{\partial u_{l,i}}\left(u_{l,i}\right) = s_{l,i}^{p-1}\frac{\partial s_{l,i}}{\partial u_{l,i}}$$

$$\simeq \begin{cases} \frac{\partial s_{l,i}}{\partial u_{l,i}} & \left(u_{l,i} \geq u_{\text{th}}\right), \\ 0 & \left(u_{l,i} < u_{\text{th}}\right), \end{cases} \tag{18}$$

where we used Eq. 2. From Eqs. 14 and 18, there is no intrinsic difference when $p > 1$; hence, we do not consider $p > 1$ options except for $p = 2$, which is commonly used in several studies (Esser et al., 2016; Pellegrini et al., 2021). However, the open problem still remains, *i.e.,* it cannot be theoretically decided which choice is better, $p = 1$ or $p > 1$. We experimentally examined it for $p = 1, 2$ as described in Sec. 4.

### 3.3.1 Differences from Other Penalty Terms

The other candidates for the penalty term are as follows:

$$\Omega_{\text{total}} = \frac{1}{p}\sum_{l=1}^{L}\sum_{i=1}^{d_l} s_{l,i}^p, \tag{19}$$

$$\Omega_{\text{balance}} = \frac{1}{p}\sum_{l=1}^{L}\sum_{i=1}^{d_l}\frac{1}{d_l}s_{l,i}^p, \tag{20}$$

where, for $p = 2$, $\Omega_{\text{total}}$ and $\Omega_{\text{balance}}$ are the same as those in Esser et al. (2016) and Pellegrini et al. (2021), respectively. However, we claim that neither can directly optimize the energy consumption because they do not have the proportional nature (Eq. 17), although they sparsify the spiking activity to some extent.

Fig. 1 and Table 1 show the discrepancy between the energy consumption metric and penalty terms of the model used in the following experiment. In these figures and table, we assumed that all the spiking neurons fired, *i.e.,* $s_{l,i} = 1 (\forall l, i)$, for the sake of simplicity. In this assumption, the ground truth is proportional to FLOPs without loss of generality. These figures and table indicate that the synaptic interaction penalty is precisely proportional to the energy consumption metric, but other penalties are not. In the next section, we will experimentally verify how this claim affects performance.

### 3.3.2 Normalization of Penalty Terms

Penalty terms are included in the objective function with their intensity parameter $\lambda$ as the coupling $\lambda\Omega_*$, where the symbol $*$ denotes "syn", "total", or "balance". For tractable treatment of the intensity parameter between various penalty terms or among models of various scales, we recommend normalizing the penalty terms by $\Omega_*(\mathbf{1})$, where $\mathbf{1}$ indicates $\boldsymbol{s} = \mathbf{1}$, *i.e.,* $s_{l,i} = 1 \forall l, i$. Note that replacing $\Omega_*$ with $\Omega_*/\Omega_*(\mathbf{1})$ is equivalent to replacing $\lambda$ with

$$\lambda' = \lambda/\Omega_*(\mathbf{1}). \tag{21}$$

Hence, we sometimes adopt the normalized notation $\lambda'$ instead of $\lambda$.

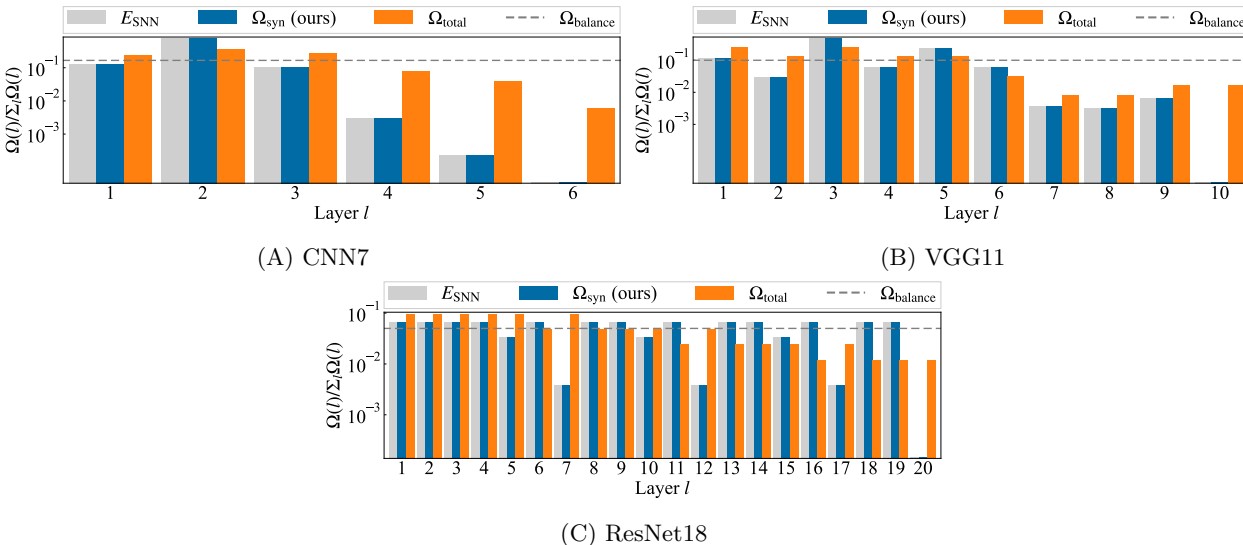

Figure 1: Comparison among layer-wise penalty terms. The $x$-axis represents the layer number, and the $y$-axis represents $E_{\mathrm{SNN}}(l)/E_{\mathrm{SNN}}$ for data of $E_{\mathrm{SNN}}$ or $\Omega_*(l)/\Omega_*$ for data of $\Omega_*$ ($*$ denotes syn, total, or balance). The network architectures are (A) CNN7, (B) VGG11, and (C) ResNet18 (App. A.1). Our penalty term (blue) is precisely proportional to the ground truth (gray).

## 4 Experiment

In this section, we evaluate the effectiveness of the proposed synaptic interaction penalty. First, we describe the setup for experiments. Next, we show that the proposed method can decrease energy consumption. Finally, we show that the proposed method can reduce the energy consumption more than other methods while maintaining the accuracy. In particular, we show that the proposed method can work under distinct surrogate gradient functions and outperforms the conversion approach (Sorbaro et al., 2020). Overall, the main objective is to analyze the behavior of our method rather than to achieve state-of-the-art accuracy.

### 4.1 Experimental Setup

As the single-step spiking neuron is developed for static tasks (Suetake et al., 2023), we experimented for the Fashion-MNIST (Xiao et al., 2017), CIFAR-10, and CIFAR-100 (Krizhevsky, 2009) datasets widely used in SNN experiments (Esser et al., 2016; Zhang & Li, 2020; Chowdhury et al., 2021b) with some network architectures, CNN7, VGG11, and ResNet18 (refer to App. A.1 for details). For these experiments, we implemented the program by the PyTorch framework and used one GPU, an NVIDIA GeForce RTX 3090, with 24 GB (refer to Table B.1 in the appendix for the difference in the training time).

In these experiments, we used the following objective function:

$$
\begin{aligned}
L = & \frac{1}{n} \sum_{n=1}^{N} \left( CE(f(x_n), t_n) + \lambda \Omega_{\mathrm{syn}}(\boldsymbol{s}(x_n)) \right) \\
& + \lambda_{\mathrm{WD}} \left( \|\boldsymbol{W}\|_{L_2}^2 + B_{\mathrm{BN}} \|\boldsymbol{W}_{\mathrm{BN}}\|_{L_2}^2 \right),
\end{aligned}
\tag{22}
$$

where $(x_n, t_n)$ denotes the pair of input data and its label, $f$ denotes some spiking neural network, $CE$ denotes the cross-entropy function, $\|\boldsymbol{W}\|_{L_2}^2$ denotes the $L_2$ penalty for the weights, *i.e.*, weight decay, $\|\boldsymbol{W}_{\mathrm{BN}}\|_{L_2}^2$ denotes the $L_2$ penalty for the trainable parameters of batch normalization layers, $\lambda$ and $\lambda_{\mathrm{WD}}$ denote the intensity of penalties, and $B_{\mathrm{BN}} \in \{0, 1\}$. Note that we included the weight decay into the objective function to verify its sparsifying effect (Yaguchi et al., 2018) in the context of SNNs. In addition, we explicitly specified the weight decay for batch normalization layers because it was included in the default setting of the PyTorch framework (Paszke et al., 2019) but not in Yaguchi et al. (2018). The training of $f$ was done by

Table 2: Performance with respect to varying $\lambda, \lambda_{\mathrm{WD}}$, and $B_{\mathrm{BN}}$. The network architecture is VGG11, the dataset is CIFAR-10, and the optimizer is Adam. $E_{\mathrm{SNN}}$ denotes the energy consumption metric for the SNN (Eq. 12), $E_{\mathrm{baseline}}$ denotes the $E_{\mathrm{SNN}}$ for the model with $\lambda = \lambda_{\mathrm{WD}} = B_{\mathrm{BN}} = 0$, dead rate denotes the ratio of the number of dead neurons to the number of total neurons, and $R$ denotes the spike firing rate (Eq. 10). All the metrics were calculated using the test dataset.

| $\lambda$ | $\lambda_{\mathrm{WD}}$ | $B_{\mathrm{BN}}$ | Accuracy [%] | $E_{\mathrm{SNN}}/E_{\mathrm{baseline}}$ [%] | Dead rate [%] | $R$ [%] |
|---|---|---|---|---|---|---|
| 0 | 0 | 0 | 89.09 ± 0.271 | 100.0 ± 1.530 | 3.367 ± 0.221 | 12.11 ± 0.114 |
| 1e-08 | 0 | 0 | 89.23 ± 0.141 | 73.92 ± 0.247 | 5.935 ± 0.822 | 10.34 ± 0.050 |
| 1e-07 | 0 | 0 | 87.81 ± 0.458 | 34.07 ± 0.397 | 22.42 ± 1.005 | 6.046 ± 0.061 |
| 1e-06 | 0 | 0 | 79.75 ± 0.694 | 11.85 ± 0.492 | 59.01 ± 1.174 | 2.176 ± 0.040 |
| 1e-05 | 0 | 0 | 21.81 ± 14.46 | 1.053 ± 1.246 | 92.43 ± 3.149 | 0.174 ± 0.198 |
| 0 | 1e-04 | 0 | 89.15 ± 0.493 | 81.07 ± 4.497 | 9.567 ± 1.005 | 8.474 ± 0.771 |
| 0 | 1e-03 | 0 | 89.00 ± 0.832 | 68.64 ± 6.714 | 18.72 ± 1.942 | 6.802 ± 0.492 |
| 0 | 1e-02 | 0 | 84.98 ± 1.455 | 44.39 ± 3.540 | 34.13 ± 0.683 | 4.992 ± 0.448 |
| 0 | 1e-01 | 0 | 50.51 ± 36.24 | 17.90 ± 16.03 | 70.93 ± 25.66 | 2.988 ± 2.642 |
| 0 | 1e+00 | 0 | 10.00 ± 0.000 | 0.000 ± 0.000 | 100.0 ± 0.000 | 0.000 ± 0.000 |
| 1e-08 | 1e-03 | 0 | 89.61 ± 0.026 | 60.28 ± 1.384 | 20.20 ± 2.268 | 6.244 ± 0.155 |
| 1e-07 | 1e-03 | 0 | 88.06 ± 0.653 | 28.72 ± 0.448 | 35.58 ± 2.255 | 4.076 ± 0.064 |
| 1e-06 | 1e-03 | 0 | 76.00 ± 0.962 | 7.829 ± 0.076 | 68.02 ± 0.504 | 1.378 ± 0.016 |
| 1e-05 | 1e-03 | 0 | 10.02 ± 0.029 | 0.007 ± 0.012 | 99.84 ± 0.283 | 0.001 ± 0.002 |
| 0 | 1e-05 | 1 | 88.99 ± 0.626 | 96.29 ± 2.003 | 6.268 ± 0.786 | 9.753 ± 0.246 |
| 0 | 1e-04 | 1 | 89.87 ± 0.526 | 85.64 ± 3.412 | 10.24 ± 1.735 | 7.457 ± 0.293 |
| 0 | 1e-03 | 1 | 86.91 ± 0.315 | 51.06 ± 1.277 | 32.79 ± 1.257 | 4.376 ± 0.138 |
| 0 | 1e-02 | 1 | 10.00 ± 0.000 | 0.000 ± 0.000 | 100.0 ± 0.000 | 0.000 ± 0.000 |
| 1e-08 | 1e-04 | 1 | 89.41 ± 0.147 | 45.87 ± 0.202 | 18.38 ± 0.631 | 5.643 ± 0.019 |
| 1e-07 | 1e-04 | 1 | 86.76 ± 0.957 | 19.21 ± 0.610 | 41.97 ± 2.169 | 2.911 ± 0.043 |
| 1e-06 | 1e-04 | 1 | 74.38 ± 0.465 | 6.582 ± 0.264 | 71.21 ± 0.608 | 1.111 ± 0.028 |
| 1e-05 | 1e-04 | 1 | 10.00 ± 0.000 | 0.003 ± 0.005 | 99.92 ± 0.140 | 0.001 ± 0.001 |

the backpropagation algorithm with the surrogate gradient of Eq. 3 unless otherwise stated. The optimizer was selected from the momentum SGD (mSGD) or Adam to confirm the claims in Yaguchi et al. (2018).

Refer to App. A for further details of the experimental setup such as hyperparameters.

## 4.2 Energy Reduction by Synaptic Interaction Penalty

We investigated whether optimizing the synaptic interaction penalty (Eq. 15) led to optimizing the energy consumption metric (Eq. 12) and whether there was any conflict with other terms in the objective function (Eq. 22). The setting was as follows. The baseline model was trained for Eq. 22 with $\lambda = \lambda_{\mathrm{WD}} = 0$. The other models were trained from scratch with some combinations of the weight decay ($\lambda_{\mathrm{WD}} > 0$), $L_2$ penalty for batch normalization layers ($B_{\mathrm{BN}} = 1$), and $p = 1$ synaptic interaction penalty ($\lambda > 0$). The results are presented in Tables 2 and 3. Note that the values in this table were taken for $\lambda$ and $\lambda_{\mathrm{WD}}$ from a point where the accuracy was very low (approximately 20%) until the accuracy reached the upper bound and stopped changing.

From the result in Table 2, we can observe the following. First, as the intensity of the penalty term increases, the energy consumption metric decreases; the inference accuracy also decreases. Therefore, the intensity parameter $\lambda$ controls the trade-off between them. Second, the combination of the synaptic interaction penalty and weight decay further reduces the energy consumption metric. Therefore, we propose to adopt both of them simultaneously. In addition, we found that the combination of the weight decay and Adam optimizer induces neuron sparsification even without the synaptic interaction penalty, though its contribution to the energy reduction is less than the synaptic interaction penalty. Furthermore, neuron sparsification proceeds more strongly, maintaining higher accuracy for the Adam optimizer than the mSGD optimizer (compare the dead rate for $\lambda = B_{\mathrm{BN}} = 0$ and $\lambda_{\mathrm{WD}} = 0$ to 1e-02 in Table 2 with that in Table 3), consistent with the findings in Yaguchi et al. (2018). Note that we cannot observe a remarkable difference between $B_{\mathrm{BN}} = 0$

Table 3: Performance with respect to varying $\lambda, \lambda_{\mathrm{WD}}$, and $B_{\mathrm{BN}}$. The optimizer is mSGD. The remaining descriptions are consistent with those detailed in Table 2.

| $\lambda$ | $\lambda_{\mathrm{WD}}$ | $B_{\mathrm{BN}}$ | Accuracy [%] | $E_{\mathrm{SNN}}/E_{\mathrm{baseline}}$ [%] | Dead rate [%] | $R$ [%] |
|---|---|---|---|---|---|---|
| 0 | 0 | 0 | $89.14 \pm 0.158$ | $100.0 \pm 2.548$ | $1.678 \pm 0.167$ | $14.19 \pm 0.277$ |
| 1e-09 | 0 | 0 | $89.36 \pm 0.159$ | $78.84 \pm 1.166$ | $2.428 \pm 0.188$ | $13.35 \pm 0.214$ |
| 1e-08 | 0 | 0 | $88.81 \pm 0.224$ | $40.00 \pm 0.335$ | $11.04 \pm 0.139$ | $10.07 \pm 0.097$ |
| 1e-07 | 0 | 0 | $86.80 \pm 0.348$ | $18.58 \pm 0.384$ | $33.62 \pm 0.875$ | $4.476 \pm 0.143$ |
| 1e-06 | 0 | 0 | $24.32 \pm 7.158$ | $73.27 \pm 98.19$ | $78.72 \pm 21.06$ | $4.207 \pm 4.165$ |
| 0 | 1e-04 | 0 | $88.59 \pm 0.413$ | $84.40 \pm 6.843$ | $2.705 \pm 0.754$ | $12.73 \pm 0.344$ |
| 0 | 1e-03 | 0 | $88.60 \pm 2.134$ | $70.96 \pm 8.041$ | $3.968 \pm 1.326$ | $11.97 \pm 0.582$ |
| 0 | 1e-02 | 0 | $70.80 \pm 2.795$ | $45.18 \pm 3.008$ | $19.57 \pm 2.636$ | $10.85 \pm 0.651$ |
| 0 | 1e-01 | 0 | $20.23 \pm 17.72$ | $18.52 \pm 13.99$ | $92.18 \pm 10.94$ | $3.322 \pm 4.056$ |
| 1e-09 | 1e-03 | 0 | $90.96 \pm 0.202$ | $64.62 \pm 0.802$ | $3.976 \pm 0.196$ | $11.82 \pm 0.085$ |
| 1e-08 | 1e-03 | 0 | $90.06 \pm 0.387$ | $35.65 \pm 0.076$ | $13.51 \pm 0.195$ | $9.043 \pm 0.068$ |
| 1e-07 | 1e-03 | 0 | $87.74 \pm 0.257$ | $16.32 \pm 0.166$ | $37.68 \pm 0.598$ | $4.163 \pm 0.040$ |
| 1e-06 | 1e-03 | 0 | $21.42 \pm 9.467$ | $26.13 \pm 24.14$ | $87.16 \pm 4.910$ | $3.255 \pm 1.576$ |
| 0 | 1e-07 | 1 | $89.30 \pm 0.263$ | $99.75 \pm 2.576$ | $1.798 \pm 0.260$ | $14.17 \pm 0.270$ |
| 0 | 1e-06 | 1 | $88.85 \pm 0.277$ | $94.59 \pm 2.186$ | $1.838 \pm 0.117$ | $13.63 \pm 0.108$ |
| 0 | 1e-05 | 1 | $88.80 \pm 0.235$ | $95.85 \pm 1.806$ | $1.834 \pm 0.134$ | $13.43 \pm 0.299$ |
| 0 | 1e-04 | 1 | $88.51 \pm 0.295$ | $87.16 \pm 1.614$ | $2.657 \pm 0.105$ | $10.12 \pm 0.151$ |
| 0 | 1e-03 | 1 | $84.47 \pm 0.757$ | $51.21 \pm 2.576$ | $16.90 \pm 0.388$ | $5.005 \pm 0.310$ |
| 0 | 1e-02 | 1 | $19.53 \pm 16.50$ | $1.786 \pm 3.093$ | $97.64 \pm 4.088$ | $0.164 \pm 0.284$ |
| 1e-09 | 1e-07 | 1 | $89.32 \pm 0.417$ | $78.35 \pm 1.694$ | $2.606 \pm 0.078$ | $13.24 \pm 0.143$ |
| 1e-08 | 1e-07 | 1 | $88.77 \pm 0.181$ | $39.70 \pm 0.844$ | $11.29 \pm 0.428$ | $10.04 \pm 0.173$ |
| 1e-07 | 1e-07 | 1 | $86.95 \pm 0.236$ | $18.60 \pm 0.285$ | $33.49 \pm 0.418$ | $4.478 \pm 0.130$ |
| 1e-06 | 1e-07 | 1 | $28.05 \pm 1.125$ | $78.52 \pm 105.4$ | $81.75 \pm 13.47$ | $4.500 \pm 4.411$ |

and 1. Therefore, we adopt the weight decay with $B_{\mathrm{BN}} = 0$ in further experiments to simplify our objective function. Finally, all the above results hold for not only VGG11 but also CNN7 and ResNet18 (see Tables 8–11 in the appendix).

### 4.3 Trade-off between Accuracy and Energy Efficiency

#### 4.3.1 Comparison Between Penalties

To examine the impact of distinct penalty terms on the energy consumption of trained models, we conducted a comparative experiment. The setting was as follows. For fair comparison, we used the $\lambda'$ notation for the intensity parameter of penalties rather than the raw $\lambda$ (see Eq. 21). The baseline model was trained for Eq. 22 with $\lambda' = B_{\mathrm{BN}} = 0$, and we tuned $\lambda_{\mathrm{WD}} > 0$ to obtain the highest accuracy. Then, the others were trained by varying $\lambda' > 0$ and by replacing $\Omega_{\mathrm{syn}}$ in Eq. 22 with $\Omega_{\mathrm{total}}$ or $\Omega_{\mathrm{balance}}$ from scratch. The results are shown in Fig. 2 (A) as $\lambda'$-parameterized curves of the energy–accuracy trade-off, where the energy consumption rate was produced as the energy consumption of each model normalized by that of the baseline model. Note that it is better for data to be located at the upper left corner in the figure. Refer to App. A.3 for the sampling of $\lambda'$. In addition, the quantitative analysis is presented in Table 4, where higher scores are better for all the metrics: area under the curve (AUC), Spearman's rank correlation coefficient (Spearman), and the mutual information (MI). Note that the argument of each metric represents a cutoff parameter, where data with lower accuracy than it are omitted. We introduced the cutoff parameter because training tended to break as the intensity parameter was increased for all the methods. Refer to App. A.4 for details of quantitative metrics.

From the result in Fig. 2 (A) and Table 4, we can observe the following. First, for each $\Omega_*$, the $p = 1$ option is apparently better than the $p = 2$ option. Therefore, we propose to adopt the $p = 1$ option. Note that this difference arises from the backward control as Eq. 18. We expect that the $p = 1$ option would substantially diminish the membrane potential below the spike firing threshold, even in cases where spike firing does not occur. Consequently, the likelihood of the membrane potential stay below the spike firing

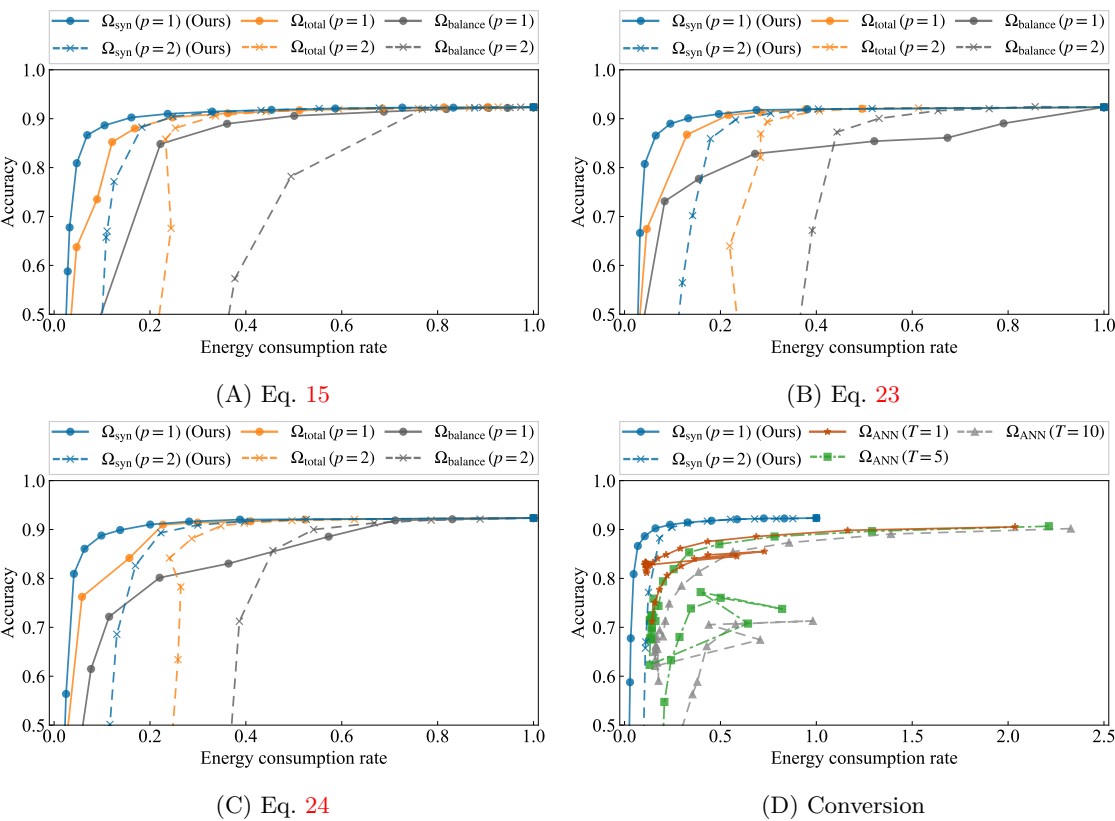

Figure 2: Energy–accuracy trade-off curves. The network architecture is CNN7, the dataset is Fashion-MNIST, and the optimizer is Adam. The energy consumption rate is the energy consumption of each model normalized by that of the baseline model. Each model is trained using the penalty term $\Omega_*$ indicated in the legend. From (A) to (C): each model is trained by the indicated surrogate gradient below each figure. (D): comparison with the conversion approach (Sorbaro et al., 2020). $\Omega_{\text{ANN}}$ denotes SNNs converted from the QReLU network. The $x$-axis exceeds one because converted SNNs have higher energy consumption than the SNN baseline for $\Omega_{\text{syn}}$.

Table 4: Quantitative comparison corresponding to Fig. 2 (A). Higher scores are better. The best and the second-best results are highlighted in bold and underlined, respectively. Refer to App. A.4 for details of the quantitative metrics.

| Method | AUC(70)[%] | AUC(50)[%] | Spearman(70) | Spearman(50) | MI(70) | MI(50) |
|---|---|---|---|---|---|---|
| $\Omega_{\text{syn}}$ ($p = 1$) (Ours) | **68.02** | **79.60** | **0.9861** | **0.9865** | **3.465** | **3.610** |
| $\Omega_{\text{syn}}$ ($p = 2$) (Ours) | 61.62 | 72.69 | 0.9474 | 0.9709 | 3.233 | 3.476 |
| $\Omega_{\text{total}}$ ($p = 1$) | 63.30 | 76.05 | 0.9766 | 0.9767 | 3.244 | 3.319 |
| $\Omega_{\text{total}}$ ($p = 2$) | 55.16 | 64.63 | 0.9831 | 0.9701 | 3.218 | 3.295 |
| $\Omega_{\text{balance}}$ ($p = 1$) | 54.23 | 67.16 | 0.9412 | 0.9412 | 2.978 | 2.978 |
| $\Omega_{\text{balance}}$ ($p = 2$) | 31.47 | 42.94 | 0.8500 | 0.8946 | 2.708 | 2.833 |

threshold following weight updates due to data fitting. However, the precise reason why the benefits of lowered energy consumption outweigh the drawbacks of potential inference accuracy deterioration remains unresolved, constituting a subject for our future investigations. Second, for $p = 1$, the trade-off curve of $\Omega_{\text{syn}}$ is the best, followed in order by $\Omega_{\text{total}}$ and $\Omega_{\text{balance}}$. Therefore, we experimentally clarified the advantage of the coefficient $\psi_{l,i}$ for Eq. 14, which had remained an issue in the method proposed by Sorbaro et al.

(2020). Finally, all the above results hold for not only CNN7 but also VGG11 and ResNet18 (see Fig. 5 and Tables 12–17 in the appendix).

### 4.3.2 Robustness to Distinct Surrogate Gradient Functions

To examine the impact of distinct functions for the surrogate gradient on the energy consumption of trained models, we conducted the same experiment as that in Sec. 4.3.1 except for the choice of a function for the surrogate gradient. Instead of Eq. 3, we adopted the piece-wise linear function (Esser et al., 2016) and scaled sigmoid (Pellegrini et al., 2021) function for the surrogate gradient as follows:

$$\frac{\partial s}{\partial u} \simeq \max\left(1 - |u - u_{\text{th}}|, 0\right), \tag{23}$$

$$\frac{\partial s}{\partial u} \simeq \frac{\partial \sigma_\alpha}{\partial u}, \tag{24}$$

where $\sigma_\alpha$ is the same as Eq. 5. The results are presented in Figs. 2 (B) and (C) (and Tables 19 and 20 in the appendix).

From the result in Figs. 2 (B) and (C), the same observations as those in Sec. 4.3.1 hold. That is, the $p = 1$ option is apparently better than the $p = 2$ option; the trade-off curve of $\Omega_{\text{syn}}$ is the best, followed in order by $\Omega_{\text{total}}$ and $\Omega_{\text{balance}}$. Therefore, the synaptic interaction penalty works under distinct surrogate gradient functions.

### 4.3.3 Superiority to Conversion Approach

To examine the impact of distinct training methods on the energy consumption of trained models, we also produced the trade-off curve for the conversion approach (Sorbaro et al., 2020). We trained a single QReLU network (an ANN with quantized ReLU activations) increasing the intensity of the penalty and evaluated the converted SNNs for each intensity in different time steps: $T = 1, 5$, and 10 (refer to the original paper (Sorbaro et al., 2020) for details). The results are presented in Fig. 2 (D) (and Table 21 in the appendix), where the energy consumption was normalized by that of the baseline for $\Omega_{\text{syn}}$.

From the result in Fig. 2 (D), we can observe that both the energy consumption and accuracy for the conversion approach are worse than those for the surrogate gradient approach. This is because the conversion process degrades the accuracy, and the penalty term for the QReLU network cannot directly optimize the energy consumption metric for the SNN. Hence, we should directly train SNNs by the surrogate gradient and synaptic interaction penalty to avoid such degradation.

### 4.3.4 Additional Trade-Off Curves

To examine the impact of distinct penalty terms on metrics beyond the energy consumption of trained models, we have included additional trade-off curves in Fig. 3 (and Tables 22–24 in the appendix). These curves depict the relationship between accuracy and specific metrics other than the energy consumption metric presented in Fig. 2 (A), while maintaining the same training procedure as adopted in Fig. 2 (A).

From the results in Fig. 3 (A, B), we can observe that training with a specific penalty term leads to a reduction in the associated metric, particularly for the $p = 1$ option compared to the $p = 2$ option. These findings underscore the suitability of the $p = 1$ option when applying a penalty term aligned with the targeted metric optimization. This choice aligns with our principle of directly optimizing the desired metric. Additionally, focusing on the results for dead neurons as displayed in Fig. 3 (C), the trade-off curve for $\Omega_{\text{syn}}$ emerges as the most favorable. Importantly, this observation serves as an advantageous outcome that complements our guiding principle. It suggests a stronger correlation between energy consumption and dead neurons compared to other metrics. Detailed analysis will be further explored in our future work.

## 5 Conclusion

We studied the training method to obtain energy-efficient SNNs in terms of the surrogate gradient. Based on our principle that we should optimize the metric as it is, we derived the synaptic interaction penalty

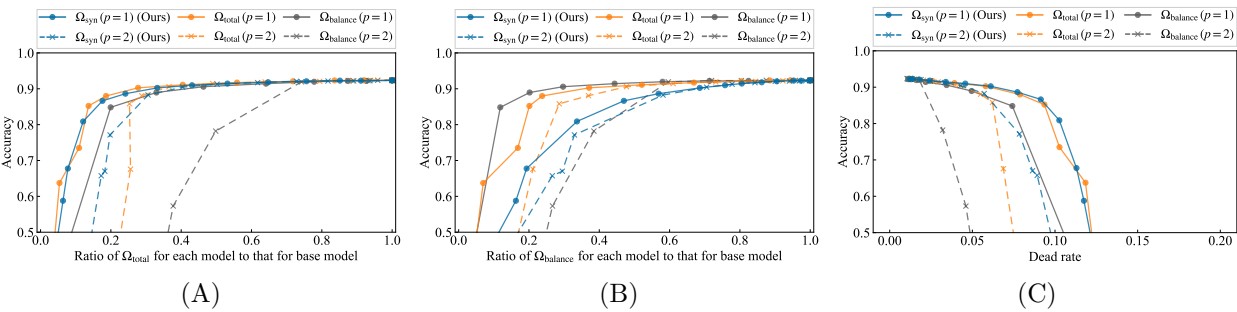

Figure 3: Additional trade-off curves corresponding to (CNN7 / Fashion-MNIST / Adam). (A), (B), and (C) indicate the trade-off curves between accuracy and the three metrics: $\Omega_{\text{total}}$, $\Omega_{\text{balance}}$, and the ratio of the number of dead neurons to the number of total neurons, respectively.

to optimize the energy consumption metric. Then, we experimentally showed that the synaptic interaction penalty (especially for $p = 1$) is superior to the existing penalties and conversion approach. Furthermore, its effectiveness remains consistent across different network architectures and choices of surrogate gradient functions. We conclude that our principle has worked well.

An apparent limitation is that the definition of the synaptic interaction penalty depends on that of the energy consumption metric. However, if the target metric becomes deformed, the penalty should be accordingly deformed in accordance with our principle—even though it is a metric irrelevant to the energy consumption. Another limitation is that although the target metric is directly included in the objective function, it is just indirectly optimized by the surrogate gradient.

We further list some outstanding issues. First, it is unclear why there was a difference between the training result for $p = 1$ and 2 of the synaptic interaction penalty. Elucidating the mechanism of this difference could help us understand the surrogate gradient. Second, we did not focus on the synergy between the spike sparsification and pruning. A pruning-aware sparsification training will help us obtain more energy-efficient SNNs. Finally, the high availability of the synaptic interaction penalty should be verified on neuromorphic chips, for example, in the case of real datasets, large networks, and other tasks. By solving these issues, we can contribute to the realization of genuinely eco-friendly SNNs.

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

Table 5: Network architectures. $\mathrm{C}(i_1)_{\mathrm{k}(i_2)\mathrm{s}(i_3)\mathrm{p}(i_4)}$ is a two-dimensional convolutional layer with channel size $= i_1$, kernel size $= i_2$, stride size $= i_3$, and padding size $= i_4$, $\mathrm{FC}(x)$ is a fully connected layer with channel size $= x$, S is a spiking activation function (Eq. 2), BN is a batch normalization layer, $\mathrm{DO}(x)$ is a dropout layer with dropout rate $= x$, Avg is a $2 \times 2$ average pooling layer, and GAP is the global average pooling layer. Both ResA and ResB are the certain residual modules—$\mathrm{ResA}(x)$ consists of two paths, [BN–S–$\mathrm{C}(x)_{\mathrm{k}(3)\mathrm{s}(1)\mathrm{p}(1)}$–BN–S–$\mathrm{C}(x)_{\mathrm{k}(3)\mathrm{s}(1)\mathrm{p}(1)}$] and the identity function; $\mathrm{ResB}(x)$ consists of two paths, [BN–S–$\mathrm{C}(x)_{\mathrm{k}(3)\mathrm{s}(2)\mathrm{p}(1)}$–BN–S–$\mathrm{C}(x)_{\mathrm{k}(3)\mathrm{s}(1)\mathrm{p}(1)}$] and [BN–S–$\mathrm{C}(x)_{\mathrm{k}(3)\mathrm{s}(1)\mathrm{p}(1)}$].

| Name | Network architecture |
|---|---|
| CNN7 | $\mathrm{C}(64)_{\mathrm{k}(3)\mathrm{s}(2)\mathrm{p}(0)}$–BN–S–DO(0.1)–$\mathrm{C}(128)_{\mathrm{k}(6)\mathrm{s}(1)\mathrm{p}(0)}$–BN–S–DO(0.2)– $\mathrm{C}(256)_{\mathrm{k}(3)\mathrm{s}(1)\mathrm{p}(0)}$–BN–S–DO(0.3)–$\mathrm{C}(128)_{\mathrm{k}(1)\mathrm{s}(1)\mathrm{p}(0)}$–BN–S–DO(0.2)– $\mathrm{C}(64)_{\mathrm{k}(1)\mathrm{s}(1)\mathrm{p}(0)}$–BN–S–DO(0.1)–$\mathrm{C}(10)_{\mathrm{k}(1)\mathrm{s}(1)\mathrm{p}(0)}$–BN–S–$\mathrm{C}(10)_{\mathrm{k}(1)\mathrm{s}(1)\mathrm{p}(0)}$–GAP |
| VGG11 | $\mathrm{C}(64)_{\mathrm{k}(3)\mathrm{s}(1)\mathrm{p}(1)}$–BN–S–DO(0.2)–$\mathrm{C}(128)_{\mathrm{k}(3)\mathrm{s}(1)\mathrm{p}(1)}$–BN–S–Avg– $\mathrm{C}(256)_{\mathrm{k}(3)\mathrm{s}(1)\mathrm{p}(1)}$–BN–S–Avg–$\mathrm{C}(512)_{\mathrm{k}(3)\mathrm{s}(1)\mathrm{p}(1)}$–BN–S–DO(0.2)– $\mathrm{C}(512)_{\mathrm{k}(3)\mathrm{s}(1)\mathrm{p}(1)}$–BN–S–Avg–$\mathrm{C}(512)_{\mathrm{k}(3)\mathrm{s}(1)\mathrm{p}(1)}$–BN–S–Avg– $\mathrm{C}(512)_{\mathrm{k}(3)\mathrm{s}(1)\mathrm{p}(1)}$–BN–S–DO(0.2)–$\mathrm{C}(512)_{\mathrm{k}(3)\mathrm{s}(1)\mathrm{p}(1)}$–BN–S– FC(4096)–BN–S–DO(0.2)–FC(4096)–BN–S–DO(0.2)–FC(10) |
| ResNet18 | $\mathrm{C}(64)_{\mathrm{k}(3)\mathrm{s}(1)\mathrm{p}(1)}$– {ResA(64)–ResA(64)}–{ResB(128)–ResA(128)– {ResB(256)–ResA(256)}–{ResB(512)–ResA(512)}– BN–S–$\mathrm{C}(10)_{\mathrm{k}(1)\mathrm{s}(1)\mathrm{p}(0)}$–GAP |

# A   Details of Experimental Setup

## A.1   Network architecture

The network architectures that we adopted in the experiments are described in Table 5. Note that $\psi_{l,i}$ is independent of $i$ in those network architectures, and all the two-dimensional convolutional and fully connected layers have no bias terms.

The batch normalization layer affects the membrane potential (Eq. 1) as follows:

$$\boldsymbol{u}_l := \frac{\boldsymbol{\alpha}_l}{\boldsymbol{\sigma}_l}\left(\boldsymbol{W}_l\boldsymbol{s}_{l-1} - \boldsymbol{\mu}_l\right) + \boldsymbol{\gamma}_l, \tag{25}$$

where $\boldsymbol{\mu}_l$ and $\boldsymbol{\sigma}_l \in \boldsymbol{R}^{d_l}$ denote the running average and standard deviation value for the post-synaptic current, $\boldsymbol{W}_l\boldsymbol{s}_{l-1}$, respectively, and $\boldsymbol{\alpha}_l$ and $\boldsymbol{\gamma}_l \in \boldsymbol{R}^{d_l}$ are the trainable affine parameters for the batch normalization layer.

## A.2   Dataset

We used three datasets as benchmarks and divided each dataset into three datasets—train, validation, and test datasets—as follows: the (#train dataset, #validation dataset, #test dataset) for each dataset is $(54000, 6000, 10000)$ for Fashion-MNIST, $(45000, 5000, 10000)$ for CIFAR-10, and $(45000, 5000, 10000)$ for CIFAR-100. We fitted trainable parameters to the training dataset, optimized hyperparameters on the validation dataset, and calculated all the metrics using the test dataset. We used random augmentation as the data augmentation technique (Cubuk et al., 2020).

## A.3   Hyperparameter

We used the following hyperparameters. The weights were initialized by He initialization for the ReLU function (although we adopted the spiking neuron) and optimized using the Adam ($\beta = (0.9, 0.999), \epsilon = 10^{-8}$) or mSGD (momentum $= 0.9$) optimizer. The mini-batch size was 100, the epoch size was 150, the spike firing threshold was $u_{\mathrm{th}} = 1$, the learning rate and $(\alpha, \tau)$ for the surrogate gradient in Eq. 3 are summarized in Table 6, and the learning rate was scheduled by the cosine annealing ($T_{\max} = 150, \eta_{\min} = 0.0$) (Loshchilov & Hutter, 2017). Note that the penalty terms $\Omega_*$ were linearly scheduled, $i.e.$, $\lambda$ was

Table 6: Hyperparameters. The following hyperparameters were used in the experiment unless otherwise stated. The learning rate, $\lambda_{\mathrm{WD}}$, $\alpha$, and $\tau$ were grid searched.

| Model | Optimizer | Surrogate gradient | Learning rate | $\lambda_{\mathrm{WD}}$ | $B_{\mathrm{BN}}$ | $\alpha$ | $\tau$ |
|---|---|---|---|---|---|---|---|
| CNN7 | Adam | Eq. 3 | 1e-3 | 1e-4 | 0 | 0.25 | 0.6 |
| CNN7 | Adam | Eq. 23 | 1e-3 | 1e-6 | 0 | - | - |
| CNN7 | Adam | Eq. 24 | 1e-2 | 1e-7 | 0 | 0.45 | - |
| CNN7 | mSGD | Eq. 3 | 1e-2 | 1e-4 | 0 | 0.35 | 0.6 |
| VGG11 | Adam | Eq. 3 | 1e-3 | 1e-3 | 0 | 0.25 | 0.6 |
| VGG11 | mSGD | Eq. 3 | 1e-2 | 1e-3 | 0 | 0.35 | 0.8 |
| ResNet18 | Adam | Eq. 3 | 1e-3 | 1e-4 | 0 | 0.35 | 1.0 |
| ResNet18 | mSGD | Eq. 3 | 1e-2 | 1e-3 | 0 | 0.35 | 1.0 |

---

**Algorithm 1** AUC($P$)

---

**Input:** data $X = \{(x_i, y_i)\}_{i=1}^N \subset \mathbf{R}_{\geq 0}^2$, cutoff $P' = P/100 \, (0 \leq P < 100)$.
$X \leftarrow \mathrm{Sort}(X, x_{i-1} \leq x_i)$;
$X \leftarrow \{(x_i, y_i) \in X \mid x_i \leq 1\}$;
$X \leftarrow (x_0 = x_1, y_0 = 0) \cup X$;
$X \leftarrow X \cup (x_\infty = 1; y_\infty = \max(\{y_i \in \mathbf{R} \mid (x_i, y_i) \in X\}))$;
$X \leftarrow \{(x_i, y_i) \in X \mid y_j \leq y_i \, (j < i)\}$;
$c \leftarrow$ linearly interpolated curve for $X$;
$A(P) \leftarrow$ Area under c over $y = P'$ for range $x \in [0, 1]$;
**Return** $A(P)/(1 - P')$.

---

multiplied by the ratio of the current epoch to the full epoch size. In the experiment for trade-off curve (Sec. 4.3), the normalized intensity parameter $\lambda'$ (Eq. 21) was selected from 14 patterns— $\{1, 2, 4, 8, 16, 32, 64, 128, 256, 512, 1024, 2048, 4096, 8192\}$. All of the experiments were conducted with three random seeds.

## A.4 Metrics

For the quantitative analysis in Sec. 4.3, we used three quantitative scores: the area under the curve (AUC), Spearman's rank correlation coefficient (Spearman), and mutual information (MI). The argument $P$ of the scores means that the score is calculated for data whose accuracy is over $P$. By AUC($P$), we mean the normalized area under the energy–accuracy trade-off curve, where the range of the $x$- and $y$-axis is $[0, 1]$ and $[P/100, 1]$, respectively. Therefore, the higher the AUC is, the lower the energy consumption metric maintaining the higher accuracy. Refer to Alg. 1 and Fig. 4 for detailed calculation of AUC($P$). Note that our algorithm of AUC overestimates the non-monotonic curves, *i.e.*, other methods than ours tend to be overestimated. Spearman's rank correlation coefficient $\rho$ describes how well two ingredients are represented by the monotonic function (Spearman, 1904). Therefore, the higher the Spearman's $\rho$ is, the higher was the aptitude of the intensity parameter as a trade-off controller. The mutual information is also suitable for the metric for a trade-off controller as it describes the mutual dependence between two ingredients, and a higher score is better.

# B Details of Experimental Result

## B.1 Difference in training time

Table 7 presents the training time ratio of each method to the baseline ($\lambda = 0$). As indicated in this table, we cannot observe the significant difference in training time. Note that a similar trend was observed in other settings.

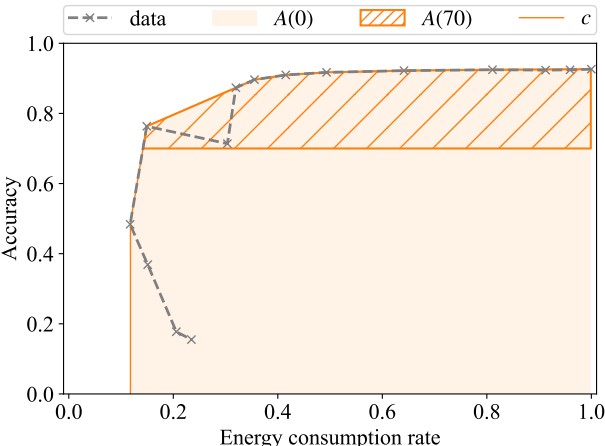

Figure 4: Example of AUC($P$) calculation (ResNet18 / CIFAR-10 / mSGD). $A(P)$ denotes an area under a curve before normalization, and $c$ denotes a linearly interpolated curve. Refer to Alg. 1 for detailed definitions of them.

Table 7: Difference in training time. The network architecture is ResNet18, the dataset is CIFAR-10, the optimizer is Adam, and $p = 1$.

| Penalty term | None | $\Omega_{\mathrm{syn}}$ | $\Omega_{\mathrm{total}}$ | $\Omega_{\mathrm{balance}}$ |
|---|---|---|---|---|
| Training time rate | 1.000 | 0.998 | 0.968 | 0.999 |

## B.2 Verification of Synaptic Interaction Penalty

The results of Sec. 4.2 are also presented in Tables 3–11. Note that the values in these tables were taken for $\lambda$ and $\lambda_{\mathrm{WD}}$ from a point where the accuracy was very low (approximately 20%) until the accuracy reached the upper bound and stopped changing.

## B.3 Energy–Accuracy Trade-Off Curve

The results of Sec. 4.3 are also presented in Fig. 5 and Tables 12–21. For all the methods, the training results of the mSGD optimizer tends to be more sensitive to the intensity of the penalty than those of the Adam optimizer. Additionally, the advantage of the proposed method over existing methods in VGG11 and ResNet18 appears to be smaller than that in CNN7. This discrepancy can be attributed to the reduced variability of $\|E_{\mathrm{SNN}}(l) - \Omega_*(l)\|$ across layers in VGG11 and ResNet18 when compared to in CNN7 (Fig. 1). A detailed analysis of this observation will be part of our future work.

## B.4 Additional Trade-Off Curves

The results from Sec. 4.3.4 are also presented in Tables 22–23.

Table 8: Performance with respect to varying $\lambda, \lambda_{\mathrm{WD}}$, and $B_{\mathrm{BN}}$. The network architecture is CNN7, the dataset is Fashion-MNIST, and the optimizer is Adam. The remaining descriptions are consistent with those detailed in Table 2.

| $\lambda$ | $\lambda_{\mathrm{WD}}$ | $B_{\mathrm{BN}}$ | Accuracy [%] | $E_{\mathrm{SNN}}/E_{\mathrm{baseline}}$ [%] | Dead rate [%] | $R$ [%] |
|---|---|---|---|---|---|---|
| 0 | 0 | 0 | 92.25 ± 0.143 | 100.0 ± 0.530 | 6.350 ± 0.055 | 14.19 ± 0.436 |
| 1e-08 | 0 | 0 | 92.26 ± 0.098 | 88.71 ± 1.472 | 6.971 ± 0.157 | 13.90 ± 0.442 |
| 1e-07 | 0 | 0 | 91.90 ± 0.120 | 49.53 ± 0.478 | 14.25 ± 0.266 | 12.33 ± 0.472 |
| 1e-06 | 0 | 0 | 90.45 ± 0.517 | 16.01 ± 0.152 | 41.91 ± 0.632 | 9.055 ± 0.272 |
| 1e-05 | 0 | 0 | 80.60 ± 4.078 | 5.289 ± 0.717 | 76.00 ± 0.992 | 4.014 ± 0.303 |
| 0 | 1e-04 | 0 | 92.35 ± 0.034 | 91.96 ± 0.749 | 7.484 ± 0.095 | 14.57 ± 0.571 |
| 0 | 1e-03 | 0 | 92.05 ± 0.132 | 76.01 ± 0.889 | 12.25 ± 0.160 | 15.34 ± 0.560 |
| 0 | 1e-02 | 0 | 90.21 ± 0.035 | 54.68 ± 0.743 | 25.81 ± 0.812 | 15.26 ± 0.372 |
| 0 | 1e-01 | 0 | 69.26 ± 10.32 | 35.26 ± 1.263 | 51.27 ± 0.807 | 13.67 ± 0.980 |
| 1e-08 | 1e-04 | 0 | 92.31 ± 0.165 | 82.88 ± 0.417 | 8.169 ± 0.084 | 14.34 ± 0.706 |
| 1e-07 | 1e-04 | 0 | 92.01 ± 0.059 | 48.29 ± 0.268 | 15.24 ± 0.379 | 13.09 ± 0.571 |
| 1e-06 | 1e-04 | 0 | 90.42 ± 0.320 | 15.80 ± 0.426 | 43.55 ± 0.802 | 10.09 ± 0.481 |
| 1e-05 | 1e-04 | 0 | 80.59 ± 1.119 | 4.133 ± 0.533 | 78.53 ± 0.325 | 4.468 ± 0.141 |
| 0 | 1e-05 | 1 | 92.35 ± 0.161 | 98.77 ± 0.450 | 6.410 ± 0.298 | 14.18 ± 0.578 |
| 0 | 1e-04 | 1 | 92.39 ± 0.138 | 89.38 ± 0.876 | 7.771 ± 0.102 | 14.70 ± 0.524 |
| 0 | 1e-03 | 1 | 91.68 ± 0.045 | 61.78 ± 1.470 | 19.44 ± 0.540 | 13.74 ± 0.391 |
| 0 | 1e-02 | 1 | 81.55 ± 0.215 | 22.13 ± 0.868 | 68.90 ± 0.260 | 8.833 ± 0.712 |
| 0 | 1e-01 | 1 | 10.00 ± 0.000 | 0.000 ± 0.000 | 100.0 ± 0.000 | 0.000 ± 0.000 |
| 1e-08 | 1e-04 | 1 | 92.52 ± 0.025 | 79.23 ± 0.457 | 8.763 ± 0.074 | 14.27 ± 0.468 |
| 1e-07 | 1e-04 | 1 | 91.86 ± 0.068 | 44.18 ± 0.395 | 17.35 ± 0.476 | 12.97 ± 0.461 |
| 1e-06 | 1e-04 | 1 | 90.35 ± 0.187 | 14.15 ± 0.315 | 46.94 ± 0.790 | 9.987 ± 0.200 |
| 1e-05 | 1e-04 | 1 | 77.68 ± 1.547 | 4.128 ± 0.824 | 78.89 ± 0.918 | 4.592 ± 1.018 |

Table 9: Performance with respect to varying $\lambda, \lambda_{\mathrm{WD}}$, and $B_{\mathrm{BN}}$. The network architecture is CNN7, the dataset is Fashion-MNIST, and the optimizer is mSGD. The remaining descriptions are consistent with those detailed in Table 2.

| $\lambda$ | $\lambda_{\mathrm{WD}}$ | $B_{\mathrm{BN}}$ | Accuracy [%] | $E_{\mathrm{SNN}}/E_{\mathrm{baseline}}$ [%] | Dead rate [%] | $R$ [%] |
|---|---|---|---|---|---|---|
| 0 | 0 | 0 | 92.02 ± 0.142 | 100.0 ± 1.288 | 11.17 ± 0.749 | 14.34 ± 0.643 |
| 1e-08 | 0 | 0 | 91.96 ± 0.072 | 94.65 ± 0.996 | 11.98 ± 0.602 | 14.28 ± 0.755 |
| 1e-07 | 0 | 0 | 91.62 ± 0.119 | 61.72 ± 2.295 | 19.16 ± 1.284 | 13.12 ± 0.677 |
| 1e-06 | 0 | 0 | 90.16 ± 0.225 | 21.72 ± 0.324 | 45.37 ± 1.028 | 10.26 ± 0.781 |
| 1e-05 | 0 | 0 | 77.14 ± 0.217 | 4.651 ± 0.270 | 79.80 ± 0.902 | 4.570 ± 0.257 |
| 0 | 1e-05 | 0 | 92.11 ± 0.114 | 98.84 ± 2.827 | 11.41 ± 0.633 | 14.44 ± 0.763 |
| 0 | 1e-04 | 0 | 92.01 ± 0.130 | 98.50 ± 0.888 | 11.90 ± 0.786 | 14.76 ± 0.774 |
| 0 | 1e-03 | 0 | 91.84 ± 0.229 | 85.42 ± 2.323 | 14.92 ± 0.359 | 15.75 ± 0.979 |
| 0 | 1e-02 | 0 | 89.70 ± 0.605 | 73.98 ± 0.855 | 22.17 ± 0.477 | 15.02 ± 0.647 |
| 0 | 1e-01 | 0 | 62.45 ± 1.900 | 19.16 ± 1.442 | 62.08 ± 1.776 | 13.46 ± 0.259 |
| 1e-08 | 1e-04 | 0 | 92.13 ± 0.168 | 91.49 ± 2.061 | 12.72 ± 0.450 | 14.62 ± 0.856 |
| 1e-07 | 1e-04 | 0 | 92.01 ± 0.135 | 61.60 ± 1.440 | 19.45 ± 0.833 | 13.56 ± 0.793 |
| 1e-06 | 1e-04 | 0 | 90.48 ± 0.420 | 21.20 ± 0.451 | 45.99 ± 1.018 | 10.84 ± 0.665 |
| 1e-05 | 1e-04 | 0 | 73.40 ± 7.835 | 4.413 ± 0.958 | 80.46 ± 3.399 | 5.385 ± 1.278 |
| 0 | 1e-05 | 1 | 92.15 ± 0.154 | 100.6 ± 1.111 | 11.38 ± 0.742 | 14.44 ± 0.712 |
| 0 | 1e-04 | 1 | 92.24 ± 0.021 | 103.0 ± 1.589 | 10.89 ± 0.781 | 14.77 ± 0.877 |
| 0 | 1e-03 | 1 | 91.64 ± 0.161 | 83.64 ± 1.326 | 16.90 ± 1.336 | 13.89 ± 1.016 |
| 0 | 1e-02 | 1 | 79.16 ± 3.171 | 30.91 ± 1.941 | 70.40 ± 1.285 | 8.008 ± 0.381 |
| 0 | 1e-01 | 1 | 10.00 ± 0.000 | 0.000 ± 0.000 | 100.0 ± 0.000 | 0.000 ± 0.000 |
| 1e-08 | 1e-05 | 1 | 91.97 ± 0.077 | 93.33 ± 1.838 | 12.43 ± 0.928 | 14.26 ± 0.624 |
| 1e-07 | 1e-05 | 1 | 91.86 ± 0.165 | 61.27 ± 1.850 | 19.44 ± 0.982 | 13.24 ± 0.634 |
| 1e-06 | 1e-05 | 1 | 90.20 ± 0.170 | 21.53 ± 0.835 | 45.55 ± 1.673 | 10.45 ± 0.608 |
| 1e-05 | 1e-05 | 1 | 73.70 ± 2.893 | 4.297 ± 0.546 | 81.37 ± 2.846 | 4.293 ± 0.441 |

Table 10: Performance with respect to varying $\lambda, \lambda_{\text{WD}}$, and $B_{\text{BN}}$. The network architecture is ResNet18, the dataset is CIFAR-10, and the optimizer is Adam. The remaining descriptions are consistent with those detailed in Table 2.

| $\lambda$ | $\lambda_{\text{WD}}$ | $B_{\text{BN}}$ | Accuracy [%] | $E_{\text{SNN}}/E_{\text{baseline}}$ [%] | Dead rate [%] | $R$ [%] |
|---|---|---|---|---|---|---|
| 0 | 0 | 0 | $91.32 \pm 0.209$ | $100.0 \pm 0.293$ | $1.074 \pm 0.033$ | $17.74 \pm 0.067$ |
| 1e-09 | 0 | 0 | $91.27 \pm 0.175$ | $93.06 \pm 0.486$ | $1.325 \pm 0.020$ | $16.75 \pm 0.023$ |
| 1e-08 | 0 | 0 | $91.40 \pm 0.293$ | $58.03 \pm 0.222$ | $3.846 \pm 0.067$ | $11.69 \pm 0.055$ |
| 1e-07 | 0 | 0 | $89.46 \pm 0.215$ | $19.13 \pm 0.247$ | $25.36 \pm 0.825$ | $4.936 \pm 0.010$ |
| 1e-06 | 0 | 0 | $78.76 \pm 0.239$ | $4.328 \pm 0.192$ | $65.38 \pm 0.283$ | $1.629 \pm 0.015$ |
| 1e-05 | 0 | 0 | $19.53 \pm 2.170$ | $0.010 \pm 0.004$ | $99.36 \pm 0.094$ | $0.040 \pm 0.008$ |
| 0 | 1e-05 | 0 | $92.15 \pm 0.032$ | $92.79 \pm 0.547$ | $2.088 \pm 0.077$ | $16.55 \pm 0.086$ |
| 0 | 1e-04 | 0 | $92.21 \pm 0.196$ | $79.67 \pm 0.721$ | $5.377 \pm 0.084$ | $14.22 \pm 0.097$ |
| 0 | 1e-03 | 0 | $91.64 \pm 0.135$ | $70.58 \pm 0.114$ | $12.08 \pm 0.185$ | $12.38 \pm 0.064$ |
| 0 | 1e-02 | 0 | $86.88 \pm 0.466$ | $50.96 \pm 1.071$ | $32.84 \pm 1.270$ | $10.59 \pm 0.099$ |
| 0 | 1e-01 | 0 | $80.94 \pm 0.264$ | $26.48 \pm 0.815$ | $58.03 \pm 1.090$ | $7.163 \pm 0.073$ |
| 0 | 1e+00 | 0 | $60.29 \pm 2.386$ | $13.99 \pm 0.264$ | $76.32 \pm 1.462$ | $5.239 \pm 0.136$ |
| 1e-09 | 1e-04 | 0 | $92.44 \pm 0.093$ | $74.74 \pm 0.310$ | $5.851 \pm 0.116$ | $13.57 \pm 0.078$ |
| 1e-08 | 1e-04 | 0 | $92.12 \pm 0.177$ | $52.82 \pm 0.206$ | $8.275 \pm 0.161$ | $10.20 \pm 0.012$ |
| 1e-07 | 1e-04 | 0 | $90.18 \pm 0.135$ | $15.27 \pm 0.075$ | $31.96 \pm 0.398$ | $3.895 \pm 0.063$ |
| 1e-06 | 1e-04 | 0 | $75.38 \pm 0.766$ | $2.385 \pm 0.449$ | $74.48 \pm 1.881$ | $1.320 \pm 0.054$ |
| 1e-05 | 1e-04 | 0 | $15.95 \pm 3.741$ | $0.002 \pm 0.001$ | $99.72 \pm 0.046$ | $0.032 \pm 0.014$ |
| 0 | 1e-05 | 1 | $91.87 \pm 0.065$ | $89.63 \pm 0.447$ | $2.467 \pm 0.022$ | $15.55 \pm 0.043$ |
| 0 | 1e-04 | 1 | $91.97 \pm 0.183$ | $66.36 \pm 0.600$ | $10.98 \pm 0.076$ | $10.32 \pm 0.079$ |
| 0 | 1e-03 | 1 | $85.93 \pm 0.244$ | $23.50 \pm 0.551$ | $54.07 \pm 0.387$ | $3.321 \pm 0.039$ |
| 0 | 1e-02 | 1 | $19.22 \pm 6.452$ | $0.574 \pm 0.040$ | $99.19 \pm 0.492$ | $0.114 \pm 0.011$ |
| 1e-09 | 1e-04 | 1 | $91.79 \pm 0.215$ | $61.58 \pm 0.537$ | $11.45 \pm 0.088$ | $9.699 \pm 0.034$ |
| 1e-08 | 1e-04 | 1 | $91.85 \pm 0.187$ | $38.43 \pm 0.512$ | $15.40 \pm 0.274$ | $6.616 \pm 0.097$ |
| 1e-07 | 1e-04 | 1 | $88.97 \pm 0.227$ | $12.29 \pm 0.292$ | $40.56 \pm 0.295$ | $2.735 \pm 0.039$ |
| 1e-06 | 1e-04 | 1 | $73.40 \pm 0.533$ | $2.153 \pm 0.115$ | $77.12 \pm 0.622$ | $0.996 \pm 0.038$ |
| 1e-05 | 1e-04 | 1 | $15.79 \pm 3.494$ | $0.001 \pm 0.001$ | $99.74 \pm 0.071$ | $0.027 \pm 0.016$ |

Table 11: Performance with respect to varying $\lambda, \lambda_{\text{WD}}$, and $B_{\text{BN}}$. The network architecture is ResNet18, the dataset is CIFAR-10, and the optimizer is mSGD. The remaining descriptions are consistent with those detailed in Table 2.

| $\lambda$ | $\lambda_{\text{WD}}$ | $B_{\text{BN}}$ | Accuracy [%] | $E_{\text{SNN}}/E_{\text{baseline}}$ [%] | Dead rate [%] | $R$ [%] |
|---|---|---|---|---|---|---|
| 0 | 0 | 0 | $90.15 \pm 0.291$ | $100.0 \pm 0.583$ | $1.188 \pm 0.060$ | $17.40 \pm 0.074$ |
| 1e-09 | 0 | 0 | $90.31 \pm 0.294$ | $95.21 \pm 0.552$ | $1.275 \pm 0.032$ | $16.70 \pm 0.086$ |
| 1e-08 | 0 | 0 | $89.83 \pm 0.092$ | $60.41 \pm 0.201$ | $3.120 \pm 0.072$ | $11.57 \pm 0.040$ |
| 1e-07 | 0 | 0 | $87.86 \pm 0.274$ | $16.01 \pm 0.067$ | $26.68 \pm 0.865$ | $4.135 \pm 0.022$ |
| 1e-06 | 0 | 0 | $54.23 \pm 3.671$ | $2.911 \pm 2.568$ | $90.89 \pm 0.931$ | $1.389 \pm 0.410$ |
| 1e-05 | 0 | 0 | $15.65 \pm 3.972$ | $0.613 \pm 0.490$ | $99.04 \pm 0.247$ | $0.176 \pm 0.034$ |
| 0 | 1e-04 | 0 | $90.56 \pm 0.098$ | $98.25 \pm 0.253$ | $1.157 \pm 0.134$ | $17.26 \pm 0.038$ |
| 0 | 1e-03 | 0 | $92.59 \pm 0.150$ | $86.46 \pm 0.818$ | $1.380 \pm 0.092$ | $15.96 \pm 0.113$ |
| 0 | 1e-02 | 0 | $91.86 \pm 0.036$ | $74.17 \pm 0.965$ | $3.932 \pm 0.277$ | $14.19 \pm 0.213$ |
| 0 | 1e-01 | 0 | $88.24 \pm 0.432$ | $61.39 \pm 2.329$ | $19.10 \pm 1.422$ | $12.66 \pm 0.274$ |
| 0 | 1e+00 | 0 | $14.65 \pm 1.956$ | $17.60 \pm 1.014$ | $54.54 \pm 2.851$ | $4.290 \pm 0.438$ |
| 1e-09 | 1e-03 | 0 | $92.44 \pm 0.268$ | $81.41 \pm 0.328$ | $1.588 \pm 0.028$ | $15.23 \pm 0.026$ |
| 1e-08 | 1e-03 | 0 | $92.48 \pm 0.124$ | $48.00 \pm 0.160$ | $4.646 \pm 0.111$ | $10.37 \pm 0.056$ |
| 1e-07 | 1e-03 | 0 | $89.05 \pm 0.245$ | $11.51 \pm 0.320$ | $43.32 \pm 0.177$ | $3.997 \pm 0.050$ |
| 1e-06 | 1e-03 | 0 | $65.07 \pm 1.377$ | $0.794 \pm 0.100$ | $91.15 \pm 0.777$ | $1.105 \pm 0.044$ |
| 1e-05 | 1e-03 | 0 | $12.85 \pm 2.377$ | $0.036 \pm 0.051$ | $99.63 \pm 0.335$ | $0.065 \pm 0.066$ |
| 0 | 1e-05 | 1 | $89.98 \pm 0.142$ | $98.41 \pm 0.691$ | $1.163 \pm 0.027$ | $17.06 \pm 0.091$ |
| 0 | 1e-04 | 1 | $90.24 \pm 0.211$ | $80.90 \pm 0.597$ | $1.886 \pm 0.058$ | $13.80 \pm 0.071$ |
| 0 | 1e-03 | 1 | $89.84 \pm 0.162$ | $32.15 \pm 0.172$ | $26.02 \pm 0.286$ | $5.147 \pm 0.030$ |
| 0 | 1e-02 | 1 | $10.04 \pm 0.064$ | $0.563 \pm 0.523$ | $99.68 \pm 0.279$ | $0.111 \pm 0.099$ |
| 1e-09 | 1e-04 | 1 | $90.17 \pm 0.192$ | $74.70 \pm 0.739$ | $2.154 \pm 0.056$ | $12.91 \pm 0.114$ |
| 1e-08 | 1e-04 | 1 | $90.08 \pm 0.071$ | $44.94 \pm 0.286$ | $5.625 \pm 0.230$ | $8.558 \pm 0.045$ |
| 1e-07 | 1e-04 | 1 | $88.17 \pm 0.151$ | $13.89 \pm 0.066$ | $31.53 \pm 0.324$ | $3.586 \pm 0.016$ |
| 1e-06 | 1e-04 | 1 | $59.17 \pm 3.370$ | $1.119 \pm 0.522$ | $91.11 \pm 0.966$ | $0.992 \pm 0.092$ |
| 1e-05 | 1e-04 | 1 | $16.01 \pm 0.654$ | $1.087 \pm 1.670$ | $99.05 \pm 1.117$ | $0.236 \pm 0.257$ |

Table 12: Quantitative comparison corresponding to (CNN7 / Fashion-MNIST / mSGD). The descriptions are consistent with those detailed in Table 4.

| Method | AUC(70)[%] | AUC(50)[%] | Spearman(70) | Spearman(50) | MI(70) | MI(50) |
|---|---|---|---|---|---|---|
| $\Omega_{\text{syn}}$ ($p = 1$) (Ours) | **65.67** | **77.18** | **0.9652** | 0.7486 | **3.355** | **3.454** |
| $\Omega_{\text{syn}}$ ($p = 2$) (Ours) | 58.19 | 70.15 | 0.9387 | 0.7794 | 3.332 | 3.434 |
| $\Omega_{\text{total}}$ ($p = 1$) | 60.90 | 74.22 | 0.9557 | 0.7705 | 3.178 | 3.295 |
| $\Omega_{\text{total}}$ ($p = 2$) | 52.73 | 66.27 | 0.9313 | **0.9562** | 3.062 | 3.233 |
| $\Omega_{\text{balance}}$ ($p = 1$) | 45.42 | 59.26 | 0.9100 | 0.9229 | 2.871 | 2.926 |
| $\Omega_{\text{balance}}$ ($p = 2$) | 28.53 | 38.09 | 0.8694 | 0.8694 | 2.813 | 2.813 |

Table 13: Quantitative comparison corresponding to (VGG11 / CIFAR-10 / Adam). The descriptions are consistent with those detailed in Table 4.

| Method | AUC(70)[%] | AUC(50)[%] | Spearman(70) | Spearman(50) | MI(70) | MI(50) |
|---|---|---|---|---|---|---|
| $\Omega_{\text{syn}}$ ($p = 1$) (Ours) | **52.56** | 68.63 | **0.9365** | **0.9497** | **3.610** | **3.688** |
| $\Omega_{\text{syn}}$ ($p = 2$) (Ours) | 46.85 | 59.73 | 0.7758 | 0.8445 | 3.536 | 3.671 |
| $\Omega_{\text{total}}$ ($p = 1$) | 52.21 | **68.69** | 0.8883 | 0.9121 | 3.583 | 3.663 |
| $\Omega_{\text{total}}$ ($p = 2$) | 47.38 | 60.58 | 0.7400 | 0.7953 | 3.506 | 3.592 |
| $\Omega_{\text{balance}}$ ($p = 1$) | 41.21 | 55.66 | 0.8770 | 0.8770 | 3.344 | 3.344 |
| $\Omega_{\text{balance}}$ ($p = 2$) | 40.68 | 53.07 | 0.8633 | 0.9006 | 3.436 | 3.556 |

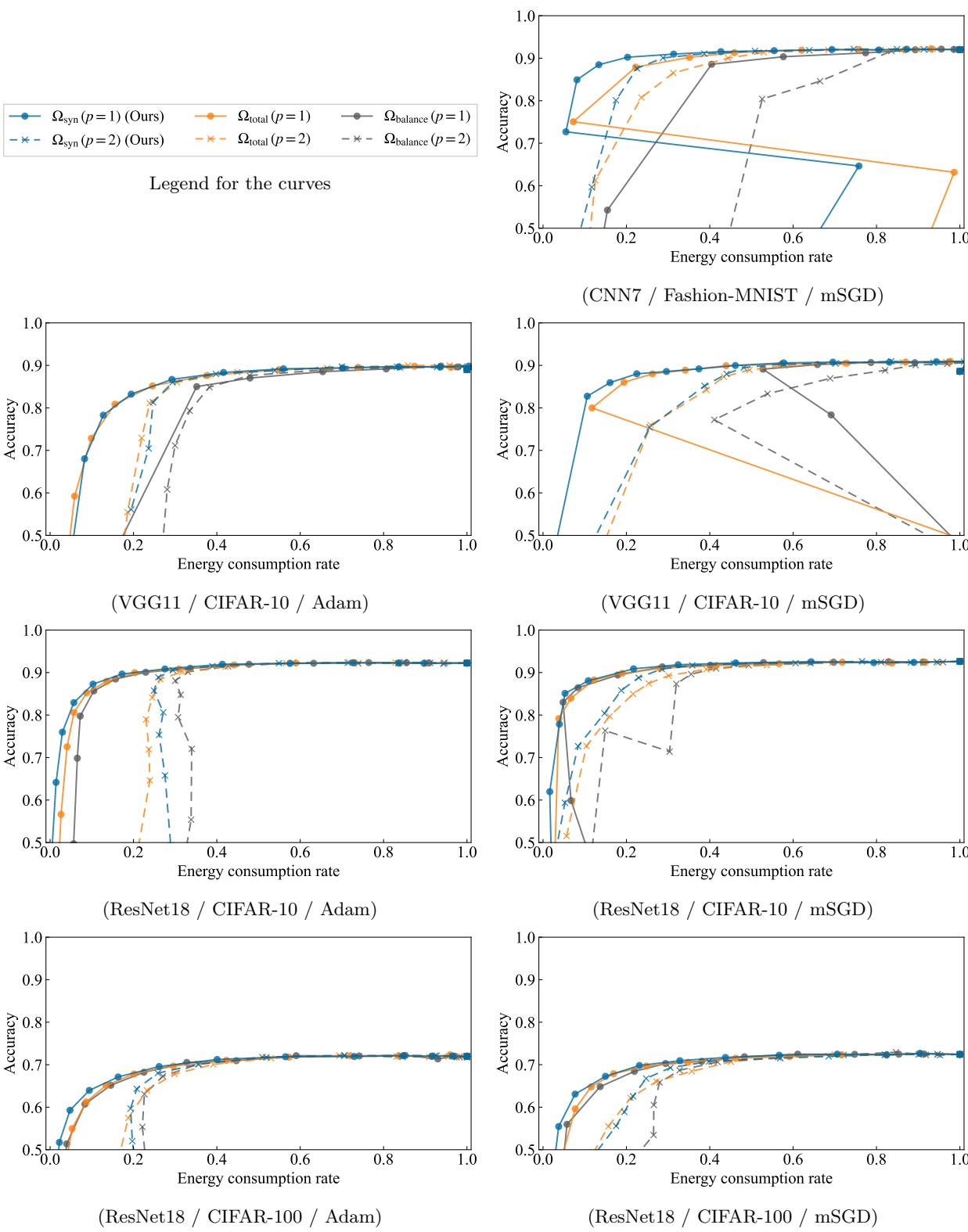

Figure 5: Energy–accuracy trade-off curve. The energy consumption rate is the energy consumption of each model normalized by that of the baseline model. The network architecture / dataset / optimizer are referred to below each figure.

Table 14: Quantitative comparison corresponding to (VGG11 / CIFAR-10 / mSGD). The descriptions are consistent with those detailed in Table 4.

| Method | AUC(70)[%] | AUC(50)[%] | Spearman(70) | Spearman(50) | MI(70) | MI(50) |
|---|---|---|---|---|---|---|
| $\Omega_{\text{syn}}$ ($p = 1$) (Ours) | **58.82** | **72.98** | 0.9576 | 0.9613 | **3.454** | **3.485** |
| $\Omega_{\text{syn}}$ ($p = 2$) (Ours) | 45.73 | 60.29 | 0.9348 | 0.9348 | 3.308 | 3.308 |
| $\Omega_{\text{total}}$ ($p = 1$) | 56.62 | 69.23 | **0.9631** | 0.9631 | 3.401 | 3.401 |
| $\Omega_{\text{total}}$ ($p = 2$) | 44.82 | 59.09 | 0.9373 | 0.9373 | 3.401 | 3.401 |
| $\Omega_{\text{balance}}$ ($p = 1$) | 35.94 | 48.02 | 0.8525 | 0.4420 | 3.075 | 3.163 |
| $\Omega_{\text{balance}}$ ($p = 2$) | 32.15 | 46.92 | 0.9591 | **0.9633** | 3.295 | 3.332 |

Table 15: Quantitative comparison corresponding to (ResNet18 / CIFAR-10 / Adam). The descriptions are consistent with those detailed in Table 4.

| Method | AUC(70)[%] | AUC(50)[%] | Spearman(70) | Spearman(50) | MI(70) | MI(50) |
|---|---|---|---|---|---|---|
| $\Omega_{\text{syn}}$ ($p = 1$) (Ours) | **67.68** | **80.07** | 0.9021 | 0.9261 | 3.308 | 3.412 |
| $\Omega_{\text{syn}}$ ($p = 2$) (Ours) | 54.60 | 62.74 | 0.8169 | 0.7917 | **3.454** | **3.573** |
| $\Omega_{\text{total}}$ ($p = 1$) | 66.51 | 78.67 | 0.9321 | 0.9475 | 3.296 | 3.400 |
| $\Omega_{\text{total}}$ ($p = 2$) | 55.08 | 64.29 | 0.8751 | 0.8846 | 3.412 | 3.536 |
| $\Omega_{\text{balance}}$ ($p = 1$) | 65.19 | 76.63 | **0.9484** | **0.9572** | 3.434 | 3.526 |
| $\Omega_{\text{balance}}$ ($p = 2$) | 51.07 | 58.70 | 0.7931 | 0.7441 | 3.412 | 3.506 |

Table 16: Quantitative comparison corresponding to (ResNet18 / CIFAR-10 / mSGD). The descriptions are consistent with those detailed in Table 4.

| Method | AUC(70)[%] | AUC(50)[%] | Spearman(70) | Spearman(50) | MI(70) | MI(50) |
|---|---|---|---|---|---|---|
| $\Omega_{\text{syn}}$ ($p = 1$) (Ours) | **68.85** | **80.56** | 0.9009 | 0.9274 | 3.244 | 3.355 |
| $\Omega_{\text{syn}}$ ($p = 2$) (Ours) | 62.15 | 75.04 | **0.9660** | **0.9730** | 3.454 | 3.545 |
| $\Omega_{\text{total}}$ ($p = 1$) | 67.47 | 79.15 | 0.9573 | 0.9573 | 3.308 | 3.308 |
| $\Omega_{\text{total}}$ ($p = 2$) | 59.02 | 72.34 | 0.9469 | 0.9552 | **3.496** | **3.555** |
| $\Omega_{\text{balance}}$ ($p = 1$) | 67.57 | 78.56 | 0.9389 | 0.9350 | 3.295 | 3.401 |
| $\Omega_{\text{balance}}$ ($p = 2$) | 56.18 | 68.44 | 0.9408 | 0.9201 | 3.319 | 3.389 |

Table 17: Quantitative comparison corresponding to (ResNet18 / CIFAR-100 / Adam). The descriptions are consistent with those detailed in Table 4. Note that the metrics with $P = 70$ are excluded because the maximum accuracy is approximately 70%.

| Method | AUC(50)[%] | Spearman(50) | MI(50) |
|---|---|---|---|
| $\Omega_{\text{syn}}$ ($p = 1$) (Ours) | **38.98** | 0.9091 | **3.496** |
| $\Omega_{\text{syn}}$ ($p = 2$) (Ours) | 33.86 | 0.8162 | 3.496 |
| $\Omega_{\text{total}}$ ($p = 1$) | 38.05 | **0.9520** | 3.454 |
| $\Omega_{\text{total}}$ ($p = 2$) | 33.15 | 0.8143 | 3.355 |
| $\Omega_{\text{balance}}$ ($p = 1$) | 38.00 | 0.8536 | 3.412 |
| $\Omega_{\text{balance}}$ ($p = 2$) | 32.69 | 0.7170 | 3.401 |

Table 18: Quantitative comparison corresponding to (ResNet18 / CIFAR-100 / mSGD). The descriptions are consistent with those detailed in Table 4. Note that the metrics associated with $P = 70$ have been excluded because the maximum accuracy is approximately 70%.

| Method | AUC(50)[%] | Spearman(50) | MI(50) |
|---|---|---|---|
| $\Omega_{\text{syn}}$ ($p = 1$) (Ours) | **39.74** | 0.9879 | **2.303** |
| $\Omega_{\text{syn}}$ ($p = 2$) (Ours) | 33.98 | 0.9704 | 2.272 |
| $\Omega_{\text{total}}$ ($p = 1$) | 38.57 | **1.0000** | **2.303** |
| $\Omega_{\text{total}}$ ($p = 2$) | 33.74 | 0.9758 | **2.303** |
| $\Omega_{\text{balance}}$ ($p = 1$) | 38.32 | 0.9636 | **2.303** |
| $\Omega_{\text{balance}}$ ($p = 2$) | 31.47 | 0.9152 | **2.303** |

Table 19: Quantitative comparison corresponding to (CNN7 / Fashion-MNIST / Adam) using Eq. 23. The descriptions are consistent with those detailed in Table 4.

| Method | AUC(70)[%] | AUC(50)[%] | Spearman(70) | Spearman(50) | MI(70) | MI(50) |
|---|---|---|---|---|---|---|
| $\Omega_{\mathrm{syn}}$ ($p=1$) (Ours) | **68.67** | **79.98** | **0.9831** | **0.9863** | **3.091** | **3.258** |
| $\Omega_{\mathrm{syn}}$ ($p=2$) (Ours) | 60.19 | 71.01 | 0.9608 | 0.9496 | 2.833 | 3.135 |
| $\Omega_{\mathrm{total}}$ ($p=1$) | 64.78 | 77.19 | 0.9765 | 0.9835 | 2.772 | 2.890 |
| $\Omega_{\mathrm{total}}$ ($p=2$) | 52.99 | 63.03 | 0.9492 | 0.9609 | 3.091 | 3.178 |
| $\Omega_{\mathrm{balance}}$ ($p=1$) | 46.25 | 65.31 | 0.7549 | 0.8211 | 2.833 | 2.944 |
| $\Omega_{\mathrm{balance}}$ ($p=2$) | 40.83 | 49.20 | 0.9588 | 0.9789 | 2.772 | 2.995 |

Table 20: Quantitative comparison corresponding to (CNN7 / Fashion-MNIST / Adam) using Eq. 24. The descriptions are consistent with those detailed in Table 4.

| Method | AUC(70)[%] | AUC(50)[%] | Spearman(70) | Spearman(50) | MI(70) | MI(50) |
|---|---|---|---|---|---|---|
| $\Omega_{\mathrm{syn}}$ ($p=1$) (Ours) | **68.66** | **80.05** | **0.9935** | **0.9948** | 3.044 | **3.178** |
| $\Omega_{\mathrm{syn}}$ ($p=2$) (Ours) | 60.04 | 71.01 | 0.9676 | 0.9797 | 2.772 | 3.091 |
| $\Omega_{\mathrm{total}}$ ($p=1$) | 64.03 | 76.76 | 0.9365 | 0.9460 | 2.813 | 2.871 |
| $\Omega_{\mathrm{total}}$ ($p=2$) | 54.16 | 62.96 | 0.9123 | 0.9123 | **3.075** | 3.075 |
| $\Omega_{\mathrm{balance}}$ ($p=1$) | 49.28 | 66.39 | 0.9236 | 0.9390 | 2.890 | 3.044 |
| $\Omega_{\mathrm{balance}}$ ($p=2$) | 39.97 | 48.82 | 0.9500 | 0.9628 | 2.772 | 2.890 |

Table 21: Quantitative comparison with Sorbaro et al. (2020) corresponding to (CNN7 / Fashion-MNIST / Adam). The descriptions are consistent with those detailed in Table 4.

| Method | AUC(70)[%] | AUC(50)[%] | Spearman(70) | Spearman(50) | MI(70) | MI(50) |
|---|---|---|---|---|---|---|
| $\Omega_{\mathrm{syn}}$ ($p=1$) (Ours) | **68.02** | **79.60** | **0.9861** | **0.9865** | 3.465 | 3.610 |
| $\Omega_{\mathrm{syn}}$ ($p=2$) (Ours) | 61.62 | 72.69 | 0.9474 | 0.9709 | 3.233 | 3.476 |
| ANN2SNN ($T=1$) | 53.51 | 67.80 | 0.6863 | 0.6275 | **4.316** | 4.355 |
| ANN2SNN ($T=5$) | 47.40 | 63.17 | 0.8548 | 0.7015 | 4.030 | 4.385 |
| ANN2SNN ($T=10$) | 37.93 | 56.30 | 0.8602 | 0.6044 | 3.784 | **4.406** |

Table 22: Quantitative comparison corresponding to Fig. 3 (A) ($\Omega_{\mathrm{total}}$). The descriptions are consistent with those detailed in Table 4.

| Method | AUC(70)[%] | AUC(50)[%] | Spearman(70) | Spearman(50) | MI(70) | MI(50) |
|---|---|---|---|---|---|---|
| $\Omega_{\mathrm{syn}}$ ($p=1$) (Ours) | 61.12 | 74.00 | 0.9879 | 0.9930 | **2.303** | **2.485** |
| $\Omega_{\mathrm{syn}}$ ($p=2$) (Ours) | 54.24 | 65.93 | **1.000** | **1.000** | 2.197 | 2.398 |
| $\Omega_{\mathrm{total}}$ ($p=1$) | **61.81** | **74.84** | **1.000** | 1.0000 | 2.197 | 2.303 |
| $\Omega_{\mathrm{total}}$ ($p=2$) | 53.48 | 63.06 | **1.000** | 0.9833 | 2.079 | 2.197 |
| $\Omega_{\mathrm{balance}}$ ($p=1$) | 55.99 | 68.73 | **1.000** | **1.000** | 1.946 | 1.946 |
| $\Omega_{\mathrm{balance}}$ ($p=2$) | 31.92 | 43.19 | 0.9000 | 0.9429 | 1.609 | 1.792 |

Table 23: Quantitative comparison corresponding to Fig. 3 (B) ($\Omega_{\mathrm{balance}}$). The descriptions are consistent with those detailed in Table 4.

| Method | AUC(70)[%] | AUC(50)[%] | Spearman(70) | Spearman(50) | MI(70) | MI(50) |
|---|---|---|---|---|---|---|
| $\Omega_{\mathrm{syn}}$ ($p=1$) (Ours) | 44.18 | 59.91 | 0.9879 | 0.9930 | **2.303** | **2.485** |
| $\Omega_{\mathrm{syn}}$ ($p=2$) (Ours) | 40.14 | 54.67 | **1.000** | **1.000** | 2.197 | 2.398 |
| $\Omega_{\mathrm{total}}$ ($p=1$) | 57.01 | 71.21 | **1.000** | 1.0000 | 2.197 | 2.303 |
| $\Omega_{\mathrm{total}}$ ($p=2$) | 51.18 | 62.91 | **1.000** | **1.000** | 2.079 | 2.197 |
| $\Omega_{\mathrm{balance}}$ ($p=1$) | **62.79** | **74.80** | **1.000** | **1.000** | 1.946 | 1.946 |
| $\Omega_{\mathrm{balance}}$ ($p=2$) | 41.15 | 53.19 | 0.9000 | 0.9429 | 1.609 | 1.792 |

Table 24: Quantitative comparison corresponding to Fig. 3 (C) (dead neurons). The descriptions are consistent with those detailed in Table 4. Note that the AUC scores have been computed based on the inverted curve relative to $x = 0.5$.

| Method | AUC(70)[%] | AUC(50)[%] | Spearman(70) | Spearman(50) | MI(70) | MI(50) |
|---|---|---|---|---|---|---|
| $\Omega_{\text{syn}}$ $(p = 1)$ (Ours) | **6.856** | **8.742** | -0.9879 | -0.9930 | **2.303** | **2.485** |
| $\Omega_{\text{syn}}$ $(p = 2)$ (Ours) | 4.919 | 6.581 | **-1.000** | **-1.000** | 2.197 | 2.398 |
| $\Omega_{\text{total}}$ $(p = 1)$ | 6.550 | 8.615 | **-1.000** | **-1.000** | 2.197 | 2.303 |
| $\Omega_{\text{total}}$ $(p = 2)$ | 4.361 | 5.432 | **-1.000** | **-1.000** | 2.079 | 2.197 |
| $\Omega_{\text{balance}}$ $(p = 1)$ | 5.122 | 6.883 | **-1.000** | **-1.000** | 1.946 | 1.946 |
| $\Omega_{\text{balance}}$ $(p = 2)$ | 2.042 | 2.939 | -0.9000 | -0.9429 | 1.609 | 1.792 |

