# OpenReview forum: "Synaptic Interaction Penalty: Appropriate Penalty Term for Energy-Efficient Spiking Neural Networks"
_TMLR — Accepted by TMLR_

### Review · Reviewer_V1jZ · 2023-10-02

**Summary Of Contributions:**

This study proposes a new penalty term to optimize the energy consumption in spiking neural networks. The author performed experiments to show the proposed penalty is better than existing penalties in terms of better accuracy given the same energy consumption.

**Audience:**

Yes

**Broader Impact Concerns:**

No.

**Claims And Evidence:**

Yes

**Requested Changes:**

### Major
I strongly suggest the author change the wording "spiking synaptic penalty". This is because the __synapse__ refers to the connection weight between neurons, then penalizing the synapse can mean some penalty on weight, e.g., the L2 penalty on weights. A potential name could be _synaptic interaction penalty_ because Eq. 13 shows the penalty is proportional to the non-negative output of the multiplication of the weight matrix with the spiking output. Some other names are OK as long as avoid this confusion.

### Minor
- Eq. 15 and 16: It looks like they don't hold but differ by the factor $T$, i.e., the size of the time step.
- The 2nd paragraph after Eq. 19: an undefined reference is shown as ??.
- A better way to illustrate the spiking synaptic penalty reflects the energy consumption metric than others is by generating a scatter plot where each dot shows the penalty and energy consumption in a layer. Then we can directly see whether the dots are in a linear line.
- it is better not to present Fig 1 on the first page because the notations are undefined at this moment and thus it is not easy to understand.

**Strengths And Weaknesses:**

This paper has a good balance of math derivations and numerical experiments, although most of the math is quite straightforward. I don't find major weaknesses in this paper whereas I think the presentation can be improved (see Requested Changes).

---

> ### Author Response · Authors · 2023-11-03
>
> Thank you for your thorough review and valuable feedback on our manuscript.
> We appreciate the time and expertise you dedicated to evaluating our work.
>
> The revised manuscript is now available, and we would be grateful if you could review it at your earliest convenience.
>
> For clarity, we have addressed each of your comments individually in the following responses:
>
> 1. > I strongly suggest the author change the wording "spiking synaptic penalty".
> * We have modified the term "spiking synaptic penalty" to "synaptic interaction penalty" based on your recommendation.
>
> 2. > Eq. 15 and 16: It looks like they don't hold but differ by the factor, i.e., the size of the time step.
> * We have introduced $T$ to the LHS of Eqs. 15 and 16 (now referred to as Eqs. 16 and 17).
>
> 3. > The 2nd paragraph after Eq. 19: an undefined reference is shown as ??.
> * We have corrected the reference error related to the figure.
>
> 4. > A better way to illustrate the spiking synaptic penalty reflects the energy consumption metric than others is by generating a scatter plot where each dot shows the penalty and energy consumption in a layer.
> * Thank you for your suggestion. However, we have decided not to adopt the scatter plot because it diminished the clarity of the relation between the ground truth and metrics compared to the previous representation.
>
> 5. > it is better not to present Fig 1 on the first page because the notations are undefined at this moment and thus it is not easy to understand.
> * We have moved Fig. 1 to Sec. 3.3.1, where notations are fully introduced, and merged with Fig. 2 (now referred to as Fig. 1).
>
> Thank you for your time and consideration.

---

> > ### Comment · Reviewer_V1jZ · 2023-12-08
> >
> > Sorry for my late reply. I have checked the authors' replies and revisions. All look good.
> > I recommend this current manuscript to be published.

---

### Review · Reviewer_kTgm · 2023-10-02

**Summary Of Contributions:**

This paper proposed a loss function to sparsify the activity of a single-step spiking neural network for energy-efficient inference. Different from traditional firing-rate-based loss functions, the proposed loss function is proportional to the energy cost of synaptic integrations in the SNN. The authors performed extensive experiments and showed that the proposed loss function could achieve better accuracy compared to firing-rate-based loss functions when having the same synaptic operation reduction. In addition, the authors also demonstrate the effect of dead neurons when using weight decay together with the sparsity-aware loss function, which further increases the energy efficiency of the network.

**Audience:**

Yes

**Broader Impact Concerns:**

There are no broader impact concerns.

**Claims And Evidence:**

Yes

**Requested Changes:**

1. Please address the concerns in the weaknesses and questions.

2.	Can the authors explain the Conv computation in Equation 6 in more detail? Does this also consider downsample operations?

3.	Does the R in Equation 9 represent the overall firing rate? It looks like R is defined as the addition of layer firing rates, but not the overall firing rate of the network.

4.	The symbols used in Figure 2 are used before definition in the main text.

5.	In the last line of Page 7, the author claimed the neuron sparsification is stronger with Adam than mSGD. However, the review cannot get the same conclusion from Tables 2 & 3. Can the authors provide more detailed interpretations of the results to support this claim?

6.	In the discussion section, the authors claim it’s possible to use equilibrium training for the proposed network. However, it’s not very clear to the reviewer how the proposed one-step network can use equilibrium training which requires the network to have a dynamical state. Can the authors give a more extended discussion on this?

**Strengths And Weaknesses:**

Strength

1.	The proposed loss function is more hardware-accurate than traditional firing-rate-based loss functions. In addition, the authors used experimental results to show the proposed loss function works as expected and gets higher energy reduction compared to firing-rate-based methods.

2.	The authors performed very detailed experiments showing many aspects of the result. The experiment includes 3 vision-based datasets and 3 convolutional neural networks. The scale of the experiment is large enough to support the claim.

Weakness

1.	The paper didn’t deliver a clear message on the dead neurons. There are mainly two important points that are not very clear:

a.	How to decide if a neuron is dead or not? Do the authors use the training set to make the decision or use validation or test set to make the decision?

b.	How does the dead neuron influence the computational cost of the network? Do the authors prune the dead neurons after training?

2.	There are some inaccurate claims in the paper:

a.	The energy cost defined in Equations 10 & 11 is not the total energy cost of SNN inference. The total energy cost of SNN inference on a typical neuromorphic processor usually consists of three components: event pre-processing cost, synaptic integration cost, and firing cost. Equation 10 & 11 only defines the synaptic integration cost. The authors need to be clear on the claim of this part.

b.	The hardware model used by the authors is under the assumption that the synaptic integration cost will dominate the energy cost of SNN inference. This is true with two conditions: first, the post-synaptic dimension needs to be high enough to balance the event pre-processing overhead; second, the activity sparsity cannot be too low. I think the model here satisfies the two conditions. However, the authors should explain these explicitly to make it clearer.

c.	The paper claimed multiple times that the proposed method is independent to surrogate gradient function. However, I think this claim is not accurate since the method needs a surrogate gradient function to work. I think it’s better to claim the method is independent to the selection of surrogate gradient function, or the method can work under different surrogate gradient functions.

Questions:

1.	The state-of-the-art ANN sparsity-aware training works normally fine-tuned trained model with sparsity loss. Why do the authors choose to train the models with sparsity loss from the beginning?

2.	From Figure 6, the reviewer observed a much smaller advantage of the proposed method than the traditional firing-rate-based method. Can the authors discuss in more detail why this happens?

---

> ### Author Response · Authors · 2023-11-03
>
> Thank you for your thorough review and valuable feedback on our manuscript.
> We appreciate the time and expertise you dedicated to evaluating our work.
>
> The revised manuscript is now available, and we would be grateful if you could review it at your earliest convenience.
>
> For clarity, we have addressed each of your comments individually in the following responses:
>
> ### Weakness
> 1. > The paper didn’t deliver a clear message on the dead neurons.
> * (For a. and b.) The dead neurons never contribute to the energy consumption metric, while they incur additional memory costs. Therefore, we prune the dead neurons to save memory capacity after training under certain conditions. However, we calculated the dead neurons using the test dataset the same as other metrics for evaluation.
> We have incorporated above discussion and information to the last paragraph of Sec. 3.2, Table 2, and A.2.
>
> 2. > There are some inaccurate claims in the paper:
> * (For a. and b.) As you pointed out, energy consumption on the chip is influenced not only by weight calculation but also by various peripheral operations. Additionally, the impact of each operation on the total energy consumption is contingent upon the chip implementation. Therefore, we have explicitly included a declaration below Def. 3.2 that our discussion is based on an ideal scenario where such peripheral operations are excluded.
> * (For c.) We have adjusted the phrasing as per your suggestion.
>
> ### Questions
> 1. > The state-of-the-art ANN sparsity-aware training works normally fine-tuned trained model with sparsity loss. Why do the authors choose to train the models with sparsity loss from the beginning?
> * As stated in A.3, we have also incorporated the linear scheduler for the proposed penalty term.
>
> 2. > From Figure 6, the reviewer observed a much smaller advantage of the proposed method than the traditional firing-rate-based method. Can the authors discuss in more detail why this happens?
> * We speculate that this phenomenon is due to the less variability of the gap between the ground truth and penalty term, leading to a diminishing discrepancy between the proposed and existing methods. We have incorporated this discussion in B.3.
>
> ### Requested Changes
> 2. > Can the authors explain the Conv computation in Equation 6 in more detail? Does this also consider downsample operations?
> * We have introduced the assertion that Eq. 6 (now referred to as Eq. 7) for convolutional computation represents an approximate value rather than the accurate value.
>
> 3. > Does the R in Equation 9 represent the overall firing rate? It looks like R is defined as the addition of layer firing rates, but not the overall firing rate of the network.
> * We apologize for any confusion. The name and definition of $R$ (Eq. 9) have been revised.
>
> 4. > The symbols used in Figure 2 are used before definition in the main text.
> * We have moved Fig. 1 to Sec. 3.3.1, where notations are fully introduced, and merged with Fig. 2 (now referred to as Fig. 1).
>
> 5. > In the last line of Page 7, the author claimed the neuron sparsification is stronger with Adam than mSGD. However, the review cannot get the same conclusion from Tables 2 & 3. Can the authors provide more detailed interpretations of the results to support this claim?
> * For instance, by comparing the dead rate for $\lambda = B_{\rm BN} = 0$ and $\lambda_{\rm WD} = 0$ to 1e-02 in Table 2 with that in Table 3, we can discern a more pronounced correlation between the dead rate and $\lambda_{\rm WD}$ in the case of Adam compared to mSGD.
> These specific points of comparison have been incorporated into the second paragraph of Sec. 4.2.
>
> 6. > In the discussion section, the authors claim it’s possible to use equilibrium training for the proposed network. However, it’s not very clear to the reviewer how the proposed one-step network can use equilibrium training which requires the network to have a dynamical state. Can the authors give a more extended discussion on this?
> * We had previously discussed equilibrium training as an approach to address the limitations of the surrogate gradient. However, we have opted to remove it from the manuscript to prevent any potential confusion.
>
> Thank you for your time and consideration.

---

### Review · Reviewer_M4zp · 2023-10-28

**Summary Of Contributions:**

This paper investigates various trade-offs between energy consumption and classification accuracy in Spiking Neural Networks (SNN), when using a loss penalty term that reduces the spike firing rate. The expected value of the chosen penalty is proportional to the SNN total energy consumption metric. The selection of the penalty term is justified theoretically. The impact of tuning the penalty term contribution to the loss is investigated, and compared against competing choices for penalty and backpropagation gradient functions. Results are reported on 3 networks (CNN7, VGG11, ResNet18) and 3 datasets (Fashion-MNIST, CIFAR-10, CIFAR-100).

**Audience:**

Yes

**Broader Impact Concerns:**

Broader impact concerns are not discussed in the manuscript. No specific concern on my end.

**Claims And Evidence:**

Yes

**Requested Changes:**

- Some minor clarifications, restructuring, and re-writing as requested under "Weaknesses"

**Strengths And Weaknesses:**

Strengths:
- the topic of optimization of SNN for reduced energy consumption is of interest for this venue
- the manuscript is, for the most part, easy to follow and understand; the theoretical derivation is especially clear
- while the use of this penalty term was previously introduced (as acknowledged in the manuscript), herein the authors justify its usage and comparatively analyze its impact on energy/accuracy trade offs
- reported results show that, subject to hyperparameter tuning, this penalty term compares favorably against alternative choices for penalty and it is effective in obtaining SSN with improved energy efficiency (according to the selected energy consumption metric) while suffering limited to no accuracy degradation
- extensive results on hyperparameter sweeps are reported and discussed
- overall, the paper presents a valuable technique to train SNN with reduced energy consumption

Weaknesses:

- The reader is immediately presented with a figure (Fig. 1) of hard interpretation. Its caption stating "the y-axis represents the ratio of some layer-wise metric or penalty term to some total metric or penalty term" is also unclear. I understand the desire to present one of the main results upfront (in this case, the proportionality between the proposed penalty term and the energy consumption) but at this early stage it is just confusing and the significance cannot be grasped until much later (Section 3.3).
- I am not sure what motivated the choice of highlighting *CCN7* results in Fig. 1, and later showing similar outcomes on VGG11 and ResNet18 in Fig. 3, while highlighting *VGG11* accuracy (Table 2 and 3), leaving CNN7 and ResNet18 results to the Appendix. Seems inconsistent
- The authors state that their objective is to discuss trends not present state-of-the-art accuracy. Although I do not disagree, starting from baselines that are 2-3% below the best available on the given networks and tasks, somewhat detracts from the strength of their conclusions.
- the proposed method requires manual selection of penalty intensity (lambda), which is model and task dependent
- Section 4.3.4 and corresponding Fig. 4 are unclear. My understanding is that the same training of previous section (Fig. 3A) is utilized, but alternative metrics are monitored and plotted against accuracy. A re-writing of the first paragraph could help the reader. In addition, when stating "training with a specific penalty term leads to a reduction in the associated metric, especially for the p=1 option", it is not clear what are these plots being compared to.

---

> ### Author Response · Authors · 2023-11-03
>
> Thank you for your thorough review and valuable feedback on our manuscript.
> We appreciate the time and expertise you dedicated to evaluating our work.
>
> The revised manuscript is now available, and we would be grateful if you could review it at your earliest convenience.
>
> For clarity, we have addressed each of your comments individually in the following responses:
>
> 1. > The reader is immediately presented with a figure (Fig. 1) of hard interpretation. Its caption stating "the y-axis represents the ratio of some layer-wise metric or penalty term to some total metric or penalty term" is also unclear. I understand the desire to present one of the main results upfront (in this case, the proportionality between the proposed penalty term and the energy consumption) but at this early stage it is just confusing and the significance cannot be grasped until much later (Section 3.3).
> * We have moved Fig. 1 to Sec. 3.3.1, where notations are fully introduced, and merged with Fig. 2 (now referred to as Fig. 1). Additionally, we have revised the description for the y-axis.
>
> 2. > I am not sure what motivated the choice of highlighting CCN7 results in Fig. 1, and later showing similar outcomes on VGG11 and ResNet18 in Fig. 3, while highlighting VGG11 accuracy (Table 2 and 3), leaving CNN7 and ResNet18 results to the Appendix. Seems inconsistent
> * As mentioned earlier, we have merged CNN7, VGG11, and ResNet18 into a single figure (Fig. 1). Due to page limitations, we had included the results of VGG11 in the main body, while the results for the other models are placed in the Appendix. Importantly, the conclusion in the main body remains consistent across all these models.
>
> 3. > The authors state that their objective is to discuss trends not present state-of-the-art accuracy. Although I do not disagree, starting from baselines that are 2-3% below the best available on the given networks and tasks, somewhat detracts from the strength of their conclusions.
> * In this study, our primary focus was on assessing the applicability of our principles in fundamental DNNs. However, as you pointed out, it remains unresolved whether our approach is also effective in recent state-of-the-art DNNs. We acknowledge this as a part of our future work and address it in the Conclusion.
>
> 4. > the proposed method requires manual selection of penalty intensity (lambda), which is model and task dependent
> * As you pointed out, the manual selection of penalty intensity is still necessary, similar to other comparison methods. However, as mentioned in the manuscript, we have improved the energy—accuracy trade-off compared to existing methods (e.g., order correlation and  mutual information), thereby making the selection process more manageable.
>
> 5. > Section 4.3.4 and corresponding Fig. 4 are unclear. My understanding is that the same training of previous section (Fig. 3A) is utilized, but alternative metrics are monitored and plotted against accuracy. A re-writing of the first paragraph could help the reader. In addition, when stating "training with a specific penalty term leads to a reduction in the associated metric, especially for the p=1 option", it is not clear what are these plots being compared to.
> * We have adjusted the wording in Sec. 4.3.4 based on your suggestion.
>
> Thank you for your time and consideration.

---

### Decision · Action_Editor_ugER · 2023-12-18

**Recommendation:** Accept as is

**Comment:**

The reviewers note that the proposed method to reduce the energy consumption of SNNs is of interest. The authors propose a loss function that is more accurate than traditional firing-rate-based loss functions. The methods is well-justified and extensive simulations support the main claims. All reviewers therefore propose acceptance of the manuscript.

**Audience:**

The topic of the paper is of interest to the TMLR audience, in particular to researchers in the field of spiking neural networks.

**Claims And Evidence:**

The claims made in the submission are sound. The authors performed extensive experiments that support the claims of the manuscript.